# A-site cation influence on the conduction band of lead bromide perovskites

Gabriel J. Man [1,8✉], Chinnathambi Kamal [2,3,4], Aleksandr Kalinko [5], Dibya Phuyal[6], Joydev Acharya [7], Soham Mukherjee[1], Pabitra K. Nayak [7], Håkan Rensmo[1], Michael Odelius [2] & Sergei M. Butorin [1✉]

Hot carrier solar cells hold promise for exceeding the Shockley-Queisser limit. Slow hot carrier cooling is one of the most intriguing properties of lead halide perovskites and distinguishes this class of materials from competing materials used in solar cells. Here we use the element selectivity of high-resolution X-ray spectroscopy and density functional theory to uncover a previously hidden feature in the conduction band states, the σ-π energy splitting, and find that it is strongly influenced by the strength of electronic coupling between the A-cation and bromide-lead sublattice. Our finding provides an alternative mechanism to the commonly discussed polaronic screening and hot phonon bottleneck carrier cooling mechanisms. Our work emphasizes the optoelectronic role of the A-cation, provides a comprehensive view of A-cation effects in the crystal and electronic structures, and outlines a broadly applicable spectroscopic approach for assessing the impact of chemical alterations of the A-cation on perovskite electronic structure.

[1] Condensed Matter Physics of Energy Materials, Division of X-ray Photon Science, Department of Physics and Astronomy, Uppsala University, Box 516 Uppsala 75121, Sweden. [2] Department of Physics, Stockholm University, AlbaNova University Center, Stockholm 10691, Sweden. [3] Theory and Simulations Laboratory, HRDS, Raja Ramanna Centre for Advanced Technology, Indore 452013, India. [4] Homi Bhabha National Institute, Training School Complex, Anushakti Nagar, Mumbai 400094, India. [5] Deutsches Elektronen-Synchrotron DESY, Notkestraße 85, Hamburg 22607, Germany. [6] Division of Material and Nano Physics, Department of Applied Physics, KTH Royal Institute of Technology, Stockholm 10691, Sweden. [7] Tata Institute of Fundamental Research, 36/P, Gopanpally Village, Serilingampally Mandal, Hyderabad 500046, India. [8] Present address: GJM Scientific Consulting, Fort Lee, New Jersey 07024, USA. ✉email: gman@alumni.princeton.edu; sergei.butorin@physics.uu.se

Lead halide perovskites (HaP) of the form $APbX_3$ have attracted renewed research interest for over a decade, motivated by initially dramatic gains in HaP solar cell efficiencies and now other optoelectronic applications[1–5]. The prototypical A-site cations (A-cations) are organic (methy-lammonium or $MA^+$, formamidinium or $FA^+$) or inorganic ($Cs^+$), the B-site cation is lead(II) and the X-site anion is iodide/bromide/chloride. At present, in spite of the substantial growth of many new subclasses of HaP-related materials and their applications, many fundamental questions related to the prototypical HaPs remain unanswered, despite a history of basic research dating back to as early as the late 1970's[6,7]. One such question concerns the optoelectronic function of the A-cation[8]. Studies based on a range of complementary approaches: time-resolved photoluminescence (PL), combined electron spectroscopy and partial density of states (PDOS) calculations, mechanical nanoindentation and solar cell characterization have yielded evidence for and against the existence of A-cation optoelectronic functionality[9–13].

To unravel this conundrum, we utilize element- and orbital-selective core level spectroscopy which includes X-ray absorption spectroscopy (XAS) for probing conduction band states and resonant X-ray emission spectroscopy (RXES) for probing valence band states[14]. The energy resolution of conventional XAS is intrinsically limited by core-hole lifetime broadening, which may obscure conduction band features probed in the spectra[15]. We circumvent this limitation by using High Energy Resolution Fluorescence Detected XAS (HERFD-XAS) which yields reduced broadenings of 2.2 eV (vs. 2.5 eV) and 2.5 eV (vs. 6.1 eV) for the bromine $K$-edge ($1s \rightarrow p$ transition) and lead $L_3$-edge ($2p_{3/2} \rightarrow s,d$ transition) absorption spectra, respectively[15,16]. Only two reports of HERFD-XAS, applied to the study of HaP thin films, exist, with just one report examining the prototypical compounds methylammonium lead tri-iodide/bromide (MAPI/B)[17,18].

In this work, we report a joint experimental and computational investigation of three prototypical lead bromide perovskites (APB): MAPB, formamidinium lead tribromide (FAPB), cesium lead tribromide (CsPB), facilitated by single crystals which are durable under X-ray irradiation. Density functional theory (DFT) was employed for ab initio molecular dynamics simulations (AIMD) combined with ground-state PDOS and bromine $K$-edge XAS calculations. The use of HERFD-XAS enables us to differentiate, in terms of electronic structure, between compounds with the same lead/bromide formal oxidation states and ultimately attribute measured differences to A-cation type. We find the strength of electronic coupling between the A-cation and bromide-lead sublattice increases in this order: $Cs^+ \rightarrow MA^+ \rightarrow FA^+$, and influences the energetic width of the conduction band via the σ–π splitting. We discuss the connection between σ–π splitting and slow cooling of hot electrons, which is relevant for potential HaP-based hot carrier solar cells which could surpass the Shockley–Queisser thermodynamic limit. We observe that higher coupling strength correlates with higher Br-Pb bond ionicity and a decreased energy offset between a given energy level and Br $1s$, and link this finding to energy level matching at optoelectronic device interfaces.

## Results

We first focus on the unoccupied conduction band states as the occupied states of HaPs have been investigated more frequently, possibly due to the higher availability of commercial photoelectron spectroscopy (PES) instrumentation. We experimentally profile the unoccupied states in an element- and orbital-resolved manner via HERFD-XAS spectra derived from RXES maps. The maps are generated with a synchrotron beamline and a crystal-based X-ray spectrometer[19,20]. A representative Br $K$ RXES map, recorded from single crystal MAPB, is displayed in Fig. 1a and features resonant and off-resonant X-ray emission from three $p \rightarrow s$ transitions: $K\beta_{1,3}$ ($3p_{3/2}/3p_{1/2} \rightarrow 1s$) and $K\beta_2$ ($4p \rightarrow 1s$). An energy level schematic is shown in Fig. 1b and depicts the one-electron transitions related to X-ray absorption into conduction band states and emission from valence band and shallow core level states. The final state of valence photo-emission is shown for reference. A representative Pb $L_3$ RXES map is displayed in Supplementary Fig. 1 and the analysis is described in Supplementary Note 1. Vertical dashed lines in all RXES maps represent constant-emission-energy map cuts through the maximum of the RXES intensity that yield the HERFD-XAS spectra. For material systems with relatively delocalized states, such a RXES map cut has been shown to be a good high-resolution approximation to the conventional XAS spectrum[21]. The conduction band states of interest are profiled in the near-edge features, which we enclose with a box marked region of interest (ROI).

The Br $K$ ROIs for $PbBr_2$, FAPB, CsPB and MAPB are displayed in Fig. 1c–f. The A-cation has an observable effect on the near-edge features. The ROI for $PbBr_2$ shows the first HERFD-XAS feature from ~13466 to ~13471 eV in incident energy. The ROIs for the APB compounds show HERFD-XAS features in the same energetic region as $PbBr_2$ and up to ~13475 eV, suggesting the higher energy part of the HERFD-XAS feature is affected by bromide interaction with the A-cation. We observe a difference in the energetic width of the HERFD-XAS feature between FAPB and CsPB, where the width of the CsPB feature is ~80% of the width of the FAPB feature. Despite the influence of the Br $1s$ core hole present in the final state of the spectroscopic process, the Br $K$ spectrum can be conceptually viewed as a reflection of the unoccupied Br $p$-state distribution. Hence we deduce that the A-cation influences the width of the conduction band and contributes states to the higher energy part of the conduction band in APB compounds. The HERFD-XAS feature at lowest photon energy is related to the part of the unoccupied states close to the conduction band minimum (CBM), relevant for optoelectronic device operation.

We plot Br $K$ HERFD-XAS cuts of $PbBr_2$, FAPB, MAPB and CsPB as one-dimensional spectra in Fig. 2a for quantitative analysis. Conventional XAS spectra recorded in Total Fluorescence Yield (TFY) mode are shown for comparison. The resolution improvement of Br $K$ HERFD-XAS (2.2 eV) versus TFY-XAS (2.5 eV) is small but is still important for resolving spectral differences between APB compounds in the energy region between ~13466 to ~13475 eV. For example, the notch in the spectrum of FAPB (~13470.3 eV, Fig. 2b), resolved with HERFD-XAS but not with TFY-XAS, indicates that the main absorption feature (main-edge) is formed of two main components. All four Br $K$ HERFD-XAS spectra show a feature at ~13469 eV. Additionally, the spectra for the APB compounds show features around 13471.5 eV which are affected by A-cation replacement, as noted above. We further confirm this deduction using calculations. The Pb $L_3$ ROIs and HERFD-XAS spectra are shown in Supplementary Fig. 1 and the analysis is described in Supplementary Note 1. We use Pb $L_3$ as a direct spectroscopic probe of changes in the electronic structure caused by A-cation-induced crystal structure changes and the resolution enhancement yielded by HERFD-XAS (2.5 vs. 6.1 eV) is crucial.

We sigmoid-fit the Br $K$ HERFD-XAS spectra of FAPB, MAPB and CsPB to extract numerical parameters (Fig. 2b). Given the signal-to-noise level of our measurements and nearly identical absorption onsets for MAPB and CsPB, we find the sigmoid fit better suited for differentiating the main-edge positions as compared to the first derivative (Supplementary Fig. 2). We find an

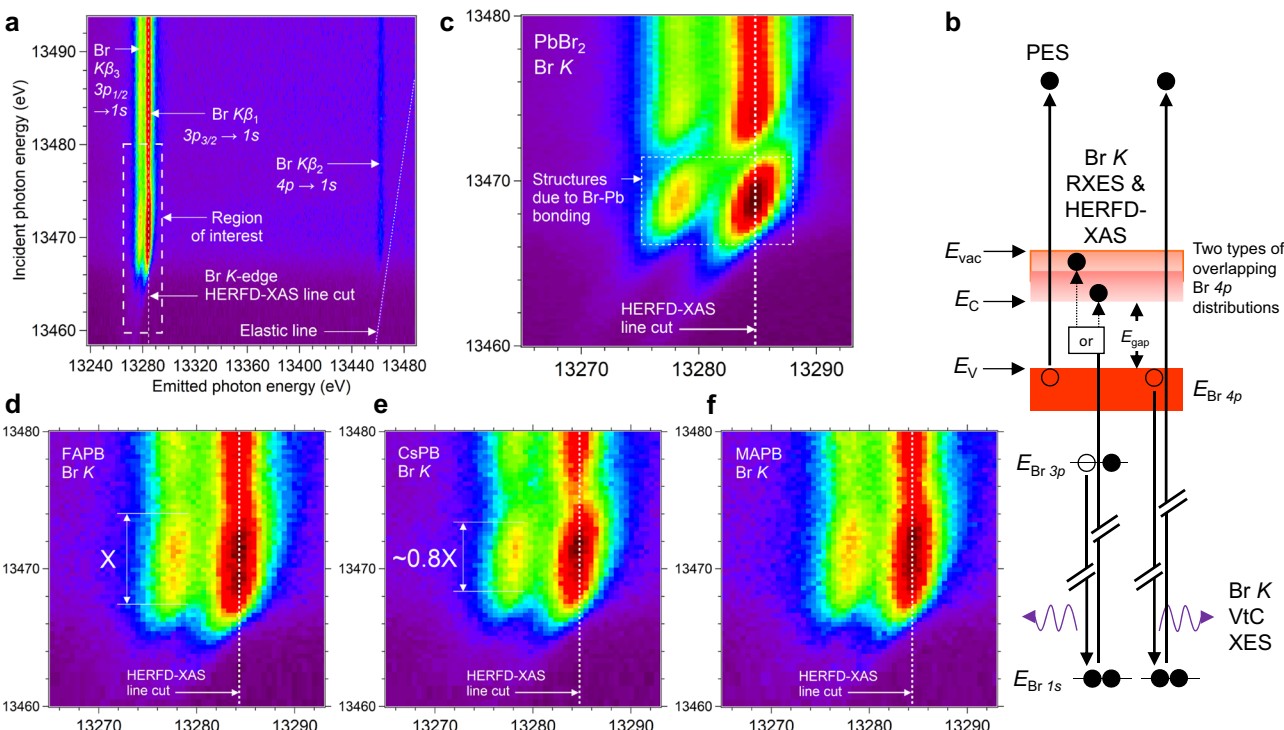

**Fig. 1 Bromine K-edge resonant XES full map and region of interest.** The High Energy Resolution Fluorescence Detected X-ray Absorption Spectroscopy (HERFD-XAS) line cuts are visualized as dashed vertical white lines in all maps. **a** Representative Br K full map, recorded from single crystal MAPB. The map shows two core-to-core transitions ($K\beta_{1,3}$) and the valence-to-core transition (VtC, $K\beta_2$), on- and off-resonance. The elastic line is marked with a guide for the eye. **b** Energy level diagram depicting the core hole decay processes measured in the Resonant X-ray Emission Spectroscopy (RXES) map, along with the valence photoemission final state for comparison. The valence and conduction band edges are labeled $E_V$ and $E_C$, respectively, $E_{gap}$ denotes the bandgap and $E_{vac}$ denotes the ionization threshold or vacuum level. **c–f** Regions of interest cut from RXES maps recorded from $PbBr_2$, FAPB, CsPB and MAPB. The x- and y-axis labels for the regions of interest are the same as the full map. Relative main-edge widths in d,e are marked with "X" and "0.8X".

absorption onset trend, based on edge positions associated with a 0.5 step height, which increases in this order: FAPB (13466.6 eV), MAPB (13467.1 eV), CsPB (13467.2 eV). The sigmoid fit uncertainty is 20 meV. The A-cation influences the absorption onsets, suggesting the A-cation affects device-relevant conduction band energy level positions referenced to Br 1s in the ground-state. The Br K spectra show a post-edge region at ~13476 eV and higher, at a sigmoid step height of 0.75. Using this step height to systematically compare main-edge widths, we estimate widths of ~7.2 eV (FAPB), ~6.9 eV (MAPB) and ~6.0 eV (CsPB).

To understand the origin of the spectral differences and link the crystal and electronic structures, we carry out periodic DFT calculations and AIMD simulations and sample calculated Br K XAS spectra using the transition potential DFT method with the half-core hole approximation[22]. To capture the structural dynamics of the A-cation accurately, we use three or four configurations, each involving between 81 and 96 bromine atoms, with 10 ps time differences in the AIMD trajectory. The longest time constant of organic cation rotation/oscillation in MAPB and FAPB has been found, via IR spectroscopy, solid-state NMR, etc. to be ~ 2 ps[23]. The 10 ps time difference covers multiple distinct rotations/oscillations, enabling us to capture the structural dynamics in a time-averaged manner. In Fig. 2c–e, we compare the calculated Br K XAS spectra, averaged over different configurations and bromide sites, against experiment. Through an analysis of the Cartesian spectral components in the crystal frame, we approximately distinguish the transitions to states of σ-symmetry in the Pb-Br-Pb direction from those of π-symmetry (as the Pb-Br-Pb is not perfectly linear) and average them separately, taking into account the crystal orientations of the Br-Pb bonds. Representative molecular orbitals for σ- and π-symmetry in MAPB are visualized in Fig. 2f.

From the XAS component spectra, we observe that the rising part of the main-edge can be assigned to Br 4p states with σ-symmetry while features in the higher energy region of the main-edge are contributed by Br 4p states with π-symmetry, in all cases. We also notice that due to the spatial extent of the states of π-symmetry, they are more strongly influenced by A-cation interactions. The lower energy position of the σ-symmetry states suggests the states in the vicinity of the CBM in the ground-state are primarily of Br-Pb σ anti-bonding character. Additional details on the comparison between experimental and calculated XAS are presented in Supplementary Note 2.

We observe a trend in the energetic separation (σ–π splitting) between the maxima of the σ- and π-symmetry distributions in the main-edge: ~3.5 eV (CsPB) → ~4.0 eV (MAPB) → ~4.2 eV (FAPB). We compare relative energy separation from calculation versus relative main-edge width from experiment. The CsPB: FAPB ratios are ~0.83 (calculation) and ~0.83 (experiment) and the MAPB: FAPB ratios are ~0.95 (calculation) and ~0.96 (experiment). We find a positive and potentially linear correlation between the relative measures of main-edge width and conclude that the A-cation influences the Br K main-edge width of the APB compounds via the σ–π splitting. The use of HERFD-XAS has revealed A-cation-induced differences in APB electronic structure, and the close agreement between experimental and calculated XAS trends offers an opportunity to computationally elucidate the σ–π splitting mechanism.

We study the Br-(A-cation) interaction using calculated Br PDOS from ground-state Kohn-Sham orbitals. For FAPB and MAPB, we choose two geometries around Br atoms, representing strong (H-Br bond distances of ~2.37 (FAPB), 2.47 Å (MAPB)) and weak (H-Br bond distance >3.0 Å) hydrogen bonds.

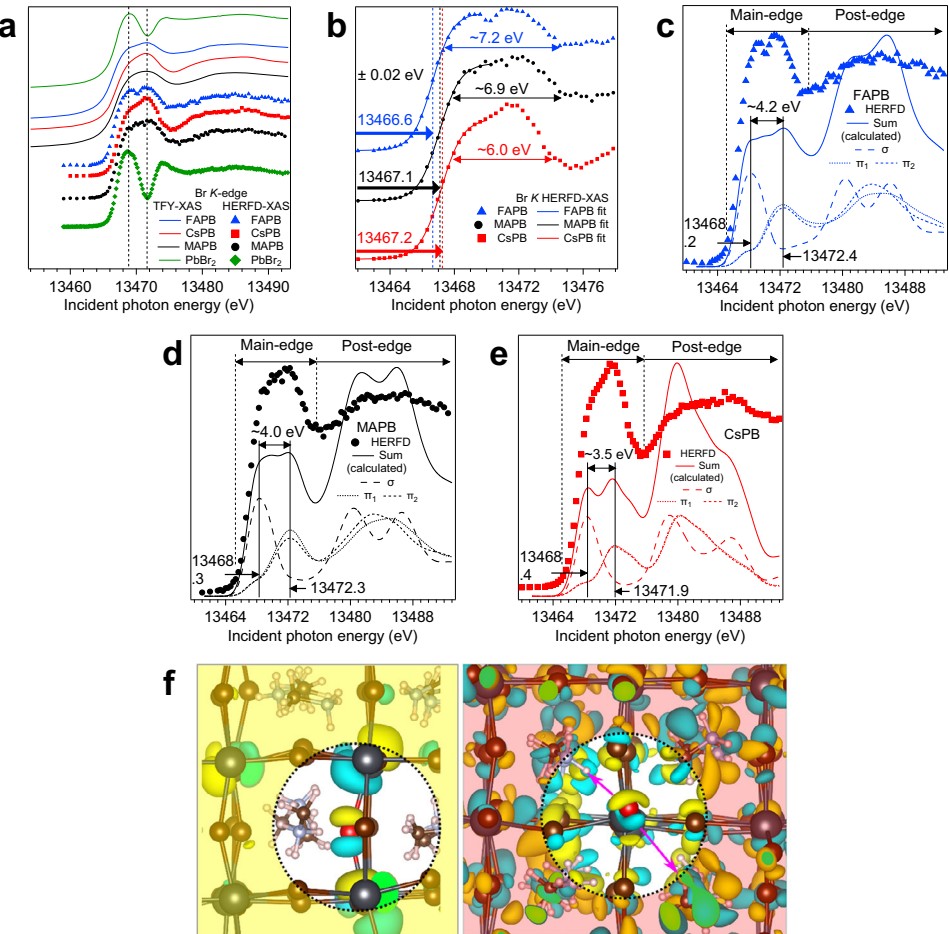

**Fig. 2 Experimental and calculated Br K-edge XAS spectra of FAPB, MAPB and CsPB. a** Comparison of the Total Fluorescence Yield X-ray absorption spectroscopy (TFY-XAS, lines) and High Energy Resolution Fluorescence Detected X-ray absorption spectroscopy (HERFD-XAS, symbols) spectra of $PbBr_2$ (green), FAPB (blue), MAPB (black) and CsPB (red). The vertical dashed lines are guides to the eye for features of interest. **b** Comparison of the HERFD-XAS main-edge spectra of the APB compounds. The absorption onsets are quantified with sigmoid fits and the main-edge widths are estimated (see text). **c–e** Experimental versus calculated Br K XAS spectra for FAPB, MAPB and CsPB. The total calculated spectrum is shown along with its constituent distributions. The sigma- and pi-symmetry distributions of states are denoted as σ, $π_1$ and $π_2$. **f** Crystal structure of MAPB, shown with the molecular orbitals associated with σ- and π-symmetry states probed with XAS. The left panel shows the σ orbitals (in cyan and yellow) emanating from the Br atom (in red). The right panel shows the π orbitals (in cyan and yellow) emanating from the Br atom (in red). The lobes of the π orbitals point towards the $MA^+$ molecule; the pointing is highlighted by the magenta arrows.

We calculate the approximate σ- and π-symmetry contributions to the Br PDOS along Pb-Br-Pb in the crystal frame (remembering there is a bend in Pb-Br-Pb) and connect them with XAS σ- and π-symmetry contributions. The Br $4s,p$ PDOS for strong hydrogen bonds is presented in Fig. 3a, b; the PDOS for weak bonds is presented in Supplementary Fig. 3. The Br $4p$ PDOS comprises the bulk of the valence band states, as shown in Supplementary Fig. 4, which displays the Br $4p$ PDOS and the total DOS (contributions from all elements). We observe that the occupied Br $4p$ π from −4 to 0.7 eV and Br $4s$ from −15 to −11.5 eV for CsPB, and MAPB/FAPB with short/long H-Br bonds are similar. The $4p$ σ from −4.5 to 0.5 eV shows some dependence on the A-cation type. On the other hand, the unoccupied Br $4p$ PDOS exhibits striking differences. The bands of π-symmetry are shifted and change shapes for FAPB or MAPB with respect to CsPB (between ~4 and ~16 eV); this clearly shows the influence of hydrogen bonding. We observe that the difference for FAPB versus CsPB is larger than for MAPB versus CsPB, in agreement with the increased hydrogen bond strength (H-Br bond distance is shorter for FAPB). We quantify the ground-state σ–π splitting with peak and center-of-gravity fits

(Supplementary Table 1) and find, irrespective of the H-Br bond distance, that the σ–π splitting increases going from CsPB → MAPB → FAPB. This is in qualitative agreement with the σ–π splitting trend in the XAS spectra.

To investigate the physicochemical interactions underlying the σ–π splitting trend, $MA^+/FA^+$ were artificially substituted with $Cs^+$ in the geometries of MAPB/FAPB while maintaining the Br–Pb sublattice structure. Remarkably, when the organic A-cations are substituted by $Cs^+$, the sharpness and energetic positions of both the π and σ contributions in the unoccupied states become CsPB-like, even though the structural models have not been relaxed. The σ–π splitting for CsPB is of comparable magnitude to the splittings for Cs-substituted MAPB/FAPB, and no systematic trend is observed for short/long H-Br bond distances (Supplementary Table 1). Hence, these artificial investigations provide computational evidence that hydrogen bonding in MAPB/FAPB has a strong influence on the conduction band and an electronic coupling exists between the A-cations and the Br-Pb sublattice. This is consistent with reported electronic coupling in the unoccupied states of MAPI[24]. Furthermore, the

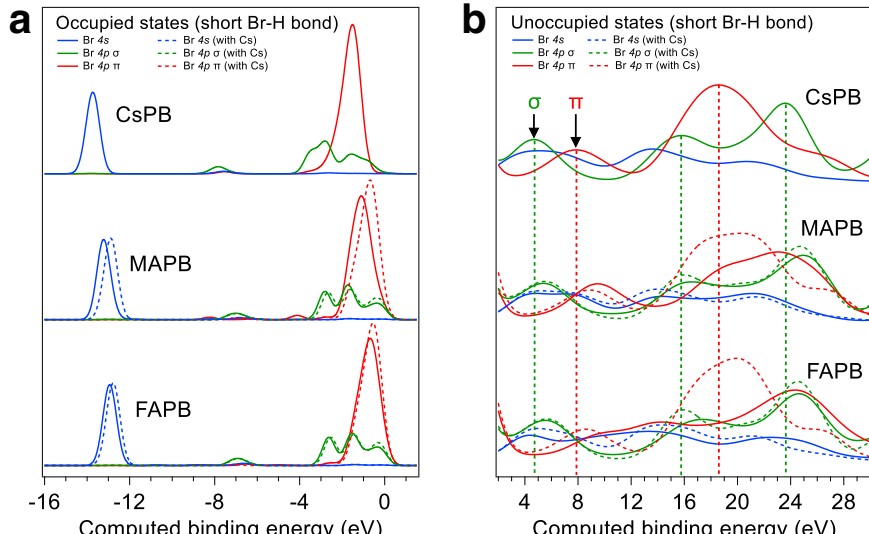

**Fig. 3 Calculated ground-state bromine 4s, 4p σ and 4p π projected density of states of CsPB and strongly hydrogen-bonded FAPB and MAPB.**
**a** Occupied states are enumerated with negative binding energies and are Gaussian-broadened by 0.3 eV. The top of the valence band is aligned to 0 eV binding energy. **b** Unoccupied states are Gaussian-broadened by 1.0 eV to emphasize the main features of interest. The two π components have been averaged together to yield one Br 4p π distribution. Projected density of states (PDOS) corresponding to strongly hydrogen-bonded bromine with organic molecules are shown. Bromine PDOS associated with cesium-substituted MAPB/FAPB are shown with dashed lines. The vertical dashed lines are guides to the eye and mark CsPB features of interest which can be compared to similar features for original and cesium-substituted MAPB and FAPB.

influence on the σ and π symmetry contributions of the Br PDOS indicates that the magnitude of the σ–π splitting in the XAS spectra is influenced by the strength of H-bonding interaction. We find a positive correlation between the average strength of N–H … Br hydrogen bonding and the average magnitude of the σ–π splitting and deduce that the rotation/oscillation of the organic A-cations likely do not have a causal relationship with the magnitude of the splitting. Further details can be found in Supplementary Discussion 1, where we also argue that the σ–π splitting is a time-averaged feature in the conduction band states and expect it to influence all electron dynamics irrespective of timescale. In Supplementary Note 3, we briefly discuss potential limitations to DFT calculations of the conduction band electronic structure, and deduce that the σ–π splitting trend is unaffected due to its magnitude of several eV.

To evaluate the possible influence of spin-orbit coupling (SOC) on the conclusions of our analysis, we performed total electronic density of states (TDOS) calculations for all three lead bromide perovskites using DFT with/without SOC effects. Further details are found in the Methods. The results of the calculations, shown in Supplementary Fig. 5, suggest that given our convolution scheme the influence of SOC on the peak positions (variations of less than 0.05 eV) at higher energy with respect to the CBM as well as the orbital characters of the states in the conduction band region is negligible. Hence, the influence of SOC on the analysis of the Br K-edge XAS data is negligible also. Furthermore, we note that the SOC effect on the lead and bromine PDOS of the three perovskites should be similar, given the same PbBr$_6$ octahedra, and in this work we focus on relative, A-cation-induced changes in the σ–π energy separation between the three perovskites. The influence of SOC is very small for the valence band region up to −10 eV, except for the Cs 5p feature in CsPB (−6 to −10 eV). The computed TDOS shows the Pb 5d splitting around −12 to −20 eV, which is consistent with the shallow core photoelectron spectra of Pb 5d (Supplementary Fig. 6) and confirms the inclusion of SOC in the TDOS calculations.

We turn our attention to possible manifestation(s) of A-cation influence in the occupied states and in the crystal structure.

Our calculations suggest the occupied Br 4p PDOS near the valence band maximum (VBM) is weakly affected by A-cation type (Fig. 3a). To investigate further, we analyze the Br K valence-to-core (VtC) XES spectra (Fig. 4 right inset) since the Br $K\beta_2$ transition, shown in the RXES map (Fig. 1a) and measured simultaneously with the $K\beta_1$ transition which yields the HERFD-XAS spectrum, carries information on the distribution of Br 4p states in the valence band. We fit a Voigt peak to the main VtC line and observe that the ~13462 to ~13472 eV region is similar for all compounds, in agreement with calculation. This finding, combined with our earlier finding of mostly Br-Pb σ-states near the CBM (Fig. 2c–e), could explain why band-edge carrier dynamics in APB crystals are independent of the A-cation type[25]. Additionally, we observe that the relative VtC intensity of FAPB is higher than MAPB/CsPB (Fig. 4 center inset), given the same number of bromide X-ray emitters (normalized to $K\beta_{1,3}$). This shows the number of electrons in Br 4p is higher for FAPB and suggests the Br-Pb bond for FAPB is more ionic. We confirm this observation by examining the chemical shift, finding an increasing Br–Pb ionicity trend (CsPB → MAPB → FAPB) which matches the σ–π splitting trend. Technical details are found in Supplementary Note 4. In principle, bond ionicity originates from the electronegativity difference between lead and bromine, hence the trend is unanticipated and reveals the A-cation influence on carrier mobility, which has previously been exponentially correlated with halide-lead bond ionicity[26]. Additionally, the $K\beta_1$ trend matches the XAS onset trend of FAPB → MAPB → CsPB, thus we see that there is a correlation between the strength of A-cation coupling and Br-Pb bond ionicity, which determines relative shifts in all energy levels referenced to Br 1s and has implications for energy level matching at device interfaces.

We utilize hard X-ray PES (HAXPES) measurements for a complementary investigation of the VBM DOS, where differences in spectral profiles could be assigned to differences in lead contributions as we have shown the Br 4p PDOS of FAPB, MAPB and CsPB are similar. We observe that the A-cation influences the DOS near the VBM (Supplementary Fig. 6) and deduce that the A-cation type modifies the degree of Pb 6s/6p hybridization with

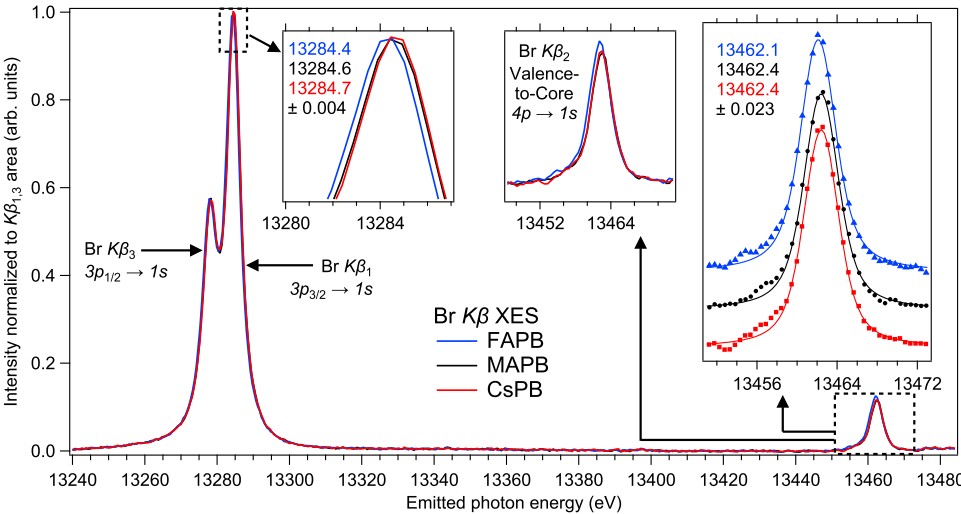

**Fig. 4 X-ray emission spectra of FAPB, MAPB and CsPB.** The $K\beta_1$ and $K\beta_2$/VtC peaks are fitted with Voigt peaks; the fitted values and uncertainties are displayed in the insets. The Voigt peak fit of the main VtC emission line is intended to extract the energy position of the peak maximum. The center inset shows VtC intensities associated with normalized $K\beta_1$ intensities.

the Br $4p$ states, which may be relevant for visible light absorption/emission and Rashba band-splitting effects. Further analysis and discussion are found in Supplementary Note 5.

Through a combined analysis of Pb $L_3$ RXES, AIMD-derived bond angles/distances (Supplementary Fig. 7) and Goldschmidt tolerance factor (GTF), we examine A-cation effects in the crystal structure[27]. Our X-ray diffraction measurements confirm the crystal structures are as expected (Supplementary Table 2) and Pb $L_3$ RXES confirms the unit cell symmetry increases with increasing GTF. We find that cooperative PbBr$_6$ octahedral tilting shows the largest geometrical variation between the APB compounds, but were unable to relate that mechanistically to the σ–π trend which instead is related directly to hydrogen bonding. We derive relative GTF ratios for MAPB: FAPB (0.98) and CsPB: FAPB (0.79) and find they nearly match the relative main-edge width MAPB: FAPB ratios of 0.96 (measured) and 0.95 (calculated) and CsPB: FAPB ratios of 0.83 (measured) and 0.83 (calculated). This is a positive and potentially linear correlation between relative GTF and relative conduction band width which applies to the APB compounds and possibly all HaP compounds. Consequently, we deduce that the GTF can be regarded as both a measure of H-bonding strength in MA$^+$/FA$^+$ (and its absence in Cs$^+$) as well as an approximate measure of octahedral tilting. Technical details can be found in Supplementary Note 6. A-cation influence on the crystal structure and optoelectronic functionality has previously been investigated[28,29]. Our XRD, Br $K$ and Pb $L_3$ HERFD-XAS, and Br $K\beta_1$ measurements show that symmetry lowering of the Br-Pb framework increases Br-Pb covalency and shifts the CBM up in potential energy, which is consistent with the findings of a pure computational study[28]. Hence, we find A-cation effects on the crystal and electronic structures that are generally consistent with previous findings. Additionally, we provide direct measurements of the relative Br–Pb bond ionicity/covalency with XES and Br $K$-edge HERFD-XAS measurements of the bromine-projected conduction band states. We summarize the numerical quantities obtained through our experimental and computational work and aforementioned correlations in Fig. 5.

**Discussion**

Using Fig. 5, we briefly discuss the implications of our findings for energy level matching at device interfaces and slow hot electron cooling. We note that direct and inverse PES measurements

of device-relevant energy level positions show a trend which systematically matches our Br $K$ XES and HERFD-XAS trends[11]. The interface energetics are sensitive to A-cation coupling strength; even a ~100 meV (~$4k_B T$ at room temperature) shift in the energy levels will substantially affect interfacial carrier transport mechanisms where current density depends exponentially on the magnitude of the energy barrier/offset (i.e. thermionic emission). State-of-the-art HaP solar cells and light-emitting diodes typically feature a mixture of MA$^+$, FA$^+$ and Cs$^+$ A-cations and depth-profiling with HAXPES has revealed spatial variations of A-cations near the surface[30,31]. Our study reveals that the A-cation(s) present, intended or not, in the few perovskite layers adjacent to the interface are expected to influence the interface energetics and hence device performance. As for the σ–π splitting, we expect it to influence slow hot carrier cooling in HaPs, which is of interest due to the technological potential of hot-carrier solar cells[32].

Single-junction solar cells, irrespective of device architecture and solar absorber material, are thermodynamically limited in power conversion efficiency (PCE) to slightly above 30%[33]. This limit assumes all excess energy from above-bandgap light absorption is unavailable to do work. Current certified PCE for thin-film crystal GaAs, single-crystal Si, and halide perovskite solar cells are 29.1%, 26.7% and 25.7%, respectively[1]. While the use of terrestrial-based, non-concentrator triple-junction solar cells (39.5% PCE) is possible, and silicon/perovskite tandem solar cells (29.8% PCE) are being commercialized, single-junction cells with PCE surpassing the Shockley-Queisser limit are highly attractive for techno-economic reasons. If the energy from above-bandgap photoexcitation is available to do work, the thermodynamic limit could be substantially higher[34]. Actual hot carrier solar cells have yet to be demonstrated, though the concept was proposed forty years ago[35].

The development of a hot carrier solar cell hinges on several prerequisites, such as the existence of slow cooling of hot carriers in the light absorber and efficient extraction of hot carriers at device interfaces[32]. At comparable electron temperatures of ~1000 K, hot carrier cooling time constants are considerably longer for HaPs (100's of ps) versus state-of-the-art GaAs (<10 ps)[12,36]. This suggests HaP-based solar cell technology has the potential to become the leading single-junction photovoltaic technology. Slow hot carrier cooling in HaPs has been experimentally observed, and phenomenologically explained

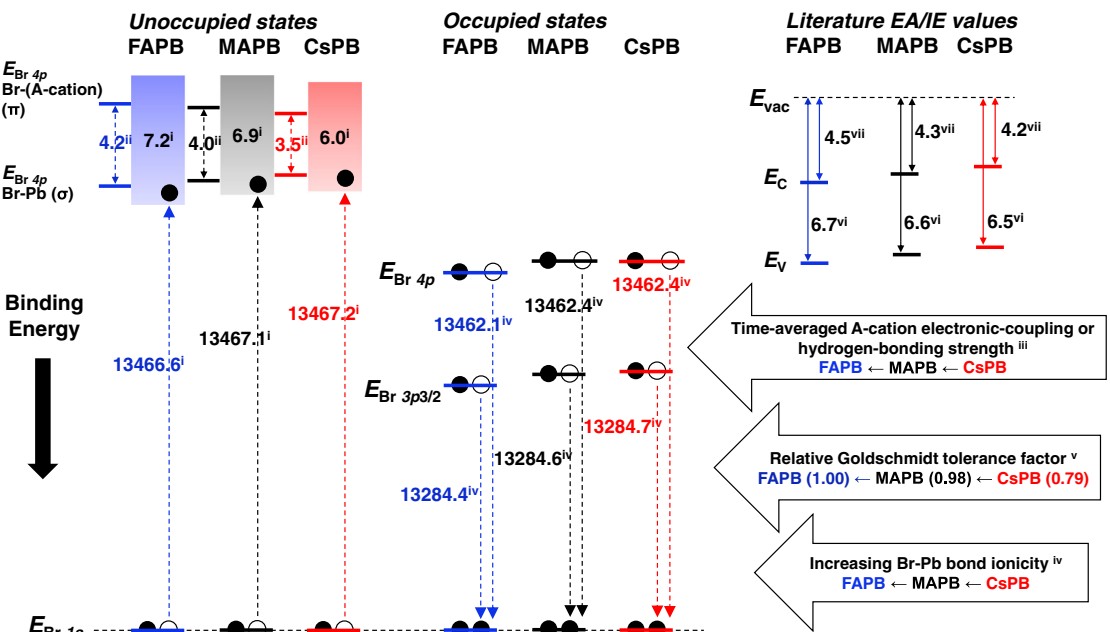

**Fig. 5 Summary of energy level positions derived from experiment and calculation and correlations found.** One-electron transitions are shown for the X-ray absorption and emission processes. The origin of each numerical value or correlation is indicated with a superscript identifier. Values obtained via experimental Bromine *K*-edge High Energy Resolution Fluorescence Detected X-ray Absorption Spectroscopy (Br *K* HERFD-XAS) measurements are identified with superscript i. Values derived from calculated Br *K* XAS are identified with superscript ii. The correlation obtained from calculated bromine projected density of states (Br PDOS) is identified with superscript iii. Values obtained via experimental Bromine *K*-edge X-ray Emission Spectroscopy (Br *K* XES) measurements are identified with superscript iv. The correlation obtained from Goldschmidt tolerance factors, calculated using parameters from the literature, is identified with superscript v. Values for the ionization energy, abbreviated as *IE*, are obtained from Ultraviolet Photoelectron Spectroscopy (UPS) measurements reported in the literature and are identified with superscript vi. Values for the electron affinity, abbreviated as *EA*, are obtained from Inverse Photoelectron Spectroscopy (IPES) measurements reported in the literature and are identified with superscript vii. A-cation electronic coupling or hydrogen-bonding strength refers to the strength of coupling between the A-cation and Br-Pb sublattice. Electron affinity and ionization energy values from the literature are shown for the purpose of comparing trends[11]. The connection between this work and the work reported in the literature is described in the text.

with the polaronic screening and hot phonon bottleneck mechanisms[12,37,38]. Our finding, of the σ–π splitting in the conduction band, provides an alternative mechanism for electron cooling, thus advancing the general understanding of electron dynamics in HaPs and spurring technological development of HaP-based hot carrier solar cells. Simultaneously, our study highlights the mostly-overlooked optoelectronic role of the A-cation, shedding light on a well-debated issue and providing guidance to HaP application developers who are tailoring HaP properties for various optoelectronic applications.

Practical HaP-based hot carrier solar cells may operate with hot electrons and/or holes, and we discuss the potential for utilizing both starting with hot holes. Slow hot hole cooling in CsPI, a related compound, has been investigated both theoretically and experimentally[39,40]. Kawai et al.[40] propose that slow hot hole cooling originates from the small DOS within 0.6 eV of the VBM and predict negligible A-cation effects on carrier cooling. Our bulk-sensitive HAXPES valence band spectra (Supplementary Fig. 6) qualitatively reveal A-cation influence on the DOS near the VBM. Since the A-cation does influence the DOS near the VBM, it is unclear if the small DOS within 0.6 eV of the VBM is responsible for overall slow hot carrier cooling. Further, high-resolution and high-statistics PES measurements are needed to quantitatively investigate A-cation influence on the DOS near the VBM.

Recently, a report which combined optical transient absorption, which measures the joint distribution of hot electrons and holes, and extreme ultraviolet (XUV) iodine 4*d*-to-5*p* transient absorption, which simultaneously and independently measures

the distributions of hot holes and holes electrons, measurements revealed that the hot electrons cool substantially slower than the hot holes in MAPI[41]. Methylammonium lead iodide is the prototypical hybrid organic-inorganic HaP and is a related compound. If slow electron cooling dominates overall slow hot carrier cooling due to faster hot hole thermalization, then electron-phonon coupling strength(s) and/or feature(s) in the conduction band states should explain the origin of overall slow hot carrier cooling. We find four pieces of experimental data in the literature that support the influence of the σ–π splitting on overall slow hot carrier cooling. First, we independently observe a ~3 eV splitting in the conduction band states from the XUV transient absorption data, referred to as a hot phonon bottleneck for electrons by Verkamp et al.[41], which persists over the entire measurement interval of picoseconds. This splitting is of comparable magnitude to the Br 4*p* splitting we find in MAPB (~4 eV), and we suggest that this feature is the σ–π splitting in the I 5*p* conduction band states of MAPI. Second, photoexcitation of single-crystal MAPB with 2.3 eV (bandgap value) yields one fluorescence decay constant of 4.4 ns while photoexcitation with above-bandgap 2.6 eV yields two decay constants: ~160 ps followed by 4.4 ns[12]. The initial ~160 ps fluorescence decay was attributed to the radiative recombination of hot carriers, though the dual decay was not interpreted on the basis of features in the electronic structure. The observation of dual decay is consistent with our finding of the energetically separated Br 4*p* σ and π PDOS in the conduction band, and also suggests the A-cation-influenced π states exhibit a higher carrier thermalization rate relative to the σ states. Third, the carrier-cooling time constant of MAPI has been reported to

increase from ~30 fs to ~30 ps as the optically-injected carrier concentration is increased from ~$10^{17}$ to ~$10^{18}$ cm$^{-3}$ [38]. The cause was attributed to the hot-phonon bottleneck mechanism, where hot phonons are not cooled quickly enough and re-heat the hot carriers. An alternative and possibly complementary explanation is that the average cooling time constant increases when a fraction of σ states are occupied upon photoexcitation, impeding fast thermalization of electrons occupying the π states. Fourth, Zhu et al.[12] have reported hot fluorescence cooling time constants, measured using picosecond time-resolved photoluminescence, for the same compounds investigated here in the same single crystal form. When photoexcited with ~3.1 eV (~0.8 eV higher than the bandgap value), Zhu et al. measure decay constants of 190 ± 20 ps (FAPB), 160 ± 10 ps (MAPB) and no measurable hot fluorescence from CsPB. We observe a positive correlation between hot fluorescence cooling time constant and σ–π splitting: ~4.2 (FAPB), ~4.0 (MAPB) and ~3.5 (CsPB) eV. Dynamical screening via solvation or large polaron formation was used to explain the long time constants. While these effects may be present during hot carrier thermalization, we suggest that a larger σ–π splitting energetically shifts the π states further away from the CBM (towards the vacuum level or ionization threshold) and thus reduces the average carrier thermalization rate.

Our finding of the σ–π splitting can explain many experimental optical spectroscopy observations reported in the literature. Taken together, our findings combined with more recent experimental evidence from the literature indicate that slow hot electron cooling dominates the overall rate of slow hot carrier cooling in HaPs, and the σ–π splitting influences slow hot electron cooling in HaPs.

Although current evidence suggests that hot holes cool much faster than hot electrons, it may still be possible to design a hot carrier HaP solar cell to extract both hot holes and electrons. Visible optical absorption depths are shorter, for shorter optical wavelengths, and the hot holes, having shorter lifetimes and hence diffusion lengths, should be photogenerated close to the sun-facing contact. A potential planar device architecture for a HaP hot carrier solar cell features the HaP-absorber/hole-selective-contact/high-workfunction electrode on the sun-facing side. This is essentially the "inverted" or "p-i-n" HaP solar cell architecture with specially designed hot carrier selective contacts. Hot electrons, having much longer lifetimes and hence diffusion lengths, could diffuse to the electron-selective contact on the backside. Properties of the carrier-selective contacts, such as energy level alignment and effective density-of-states matching, would need to be designed to pass through hot carriers and perhaps some colder carriers, to balance electron/hole extraction, photovoltage versus photocurrent, etc.[42]. At present, the "conventional" or "n-i-p" architecture, with the electron-selective-contact on the sun-facing side, has shown higher PCE (25.6%)[43] compared to the p-i-n architecture (23.7%)[44]. Aside from historical differences in development time and suboptimal cell design and/or fabrication, suboptimal selection/implementation of selective contacts, etc. in the p-i-n architecture, we cannot think of any other (intrinsic) reasons for the disparity in performance.

Aside from its influence on electron thermalization, we expect the σ–π splitting to affect electron transport. The potential connection between the σ–π character of the conduction band and polaron-like carrier transport is discussed in Supplementary Discussion 2.

In conclusion, through a joint experimental-computational investigation, we show that the energetic width of the conduction band depends on the strength of electronic coupling between the A-cation and the lead-bromide sublattice. Our finding of the σ–π splitting in the conduction band provides an additional

mechanism to consider, in the search for the origins of slow hot carrier cooling and polaronic transport. In both the occupied and unoccupied states, we find a correlation between higher coupling strength and higher bromide-lead bond ionicity and a decreased energy offset between a given energy level and Br 1s, and discuss the implications for energy level matching at perovskite device interfaces. We foresee the spectroscopic approach we have employed in this work being utilized to study A-cation effects in other halide perovskites and potentially non-halide perovskites as well, thus unraveling unexplored structure-property relationships in these materials and furthering the development of optoelectronic devices.

## Methods

**APB crystal growth.** The MAPB and FAPB single crystals were solution-grown using methods reported by Dr. Pabitra Nayak previously[45,46]. The CsPB single crystals were solution-grown using a method reported by others[47]. All single crystals were grown in an ambient air environment, then transferred into vials of chlorobenzene for storage and preservation.

**X-ray diffraction.** X-ray diffraction measurements were performed on FAPB and CsPB single crystals at room temperature using a Rigaku diffractometer with graphite-monochromated molybdenum $K\alpha$ radiation (λ = 0.71073 Å). Data processing was performed with the CrysAlisPro software. Empirical absorption correction was applied to the collected reflections with SCALE3 ABSPACK and integrated with CrysAlisPro. The structures of the single crystals were solved by direct methods using the SHELXT program, and refined with the full-matrix least-squares method based on F$^2$ by using the SHELXL program through the Olex interface. The crystallographic parameters for MAPB are taken from reference[45], since a similar growth procedure was used by the same preparer.

**Hard X-ray absorption and emission spectroscopy.** The measurements were performed (on APB crystals from the same batch characterized with XRD) at beamline P64 of the synchrotron facility PETRA III[48]. Incident photon energy calibration at the Pb $L_3$ edge was performed with metallic lead foil. The photon flux incident on the sample at ~13 keV was ~3 × $10^{11}$ photons s$^{-1}$. The X-ray spot size, measured with the X-ray eye, is ~220 μm × 100 μm. Total FY-XAS measurements were recorded with a passivated implanted planar silicon (PIPS) detector (Canberra). Resonant XES maps were recorded using a von Hamos-type hard X-ray crystal spectrometer mounted in a Bragg scattering configuration. The spectrometer featured 8 crystal analyzers; the third-order reflection of the Si(220) crystal analyzers was used for Pb $L\alpha_{1,2}$ measurements and the fourth-order reflection was used for Br $K\beta_{1,2,3}$ measurements[20]. Energy calibration of the two-dimensional detector images was performed with custom Python-based software written by Dr. Aleksandr Kalinko. The Br $K$ maps were energy-calibrated using several elastic lines which span the spectrometer energy window, while the Pb $L_3$ maps were energy-calibrated by setting the Pb $L\alpha_{1,2}$ emission energies to standard values of 10551 and 10449 eV, respectively. Bromine $K$ (~13.47 keV) and lead $L_3$ (~13.04 keV) HERFD-XAS spectra were generated as averaged intensities of the slice cut from RXES maps. The energetic width of the slice on the emitted energy axis (HERFD linewidth) was 1.6 eV and the cut was done through the RXES maximum. The HERFD linewidth was selected based on a balance between energy resolution and signal-to-noise (Supplementary Fig. 8). The off-resonant, higher incident energy half of the RXES map was used to generate the X-ray emission spectrum. The energy resolution of XES is determined by the ~0.3 eV photon bandwidth of the Si(311) monochromator and the ~1.5 eV FWHM Gaussian broadening of the spectrometer. Single crystals were prepared for measurements by first extracting them from chlorobenzene, blowdrying with compressed air/nitrogen in an ambient air environment, then mounting them into a Linkam T95 heating/cooling stage (~ 5 min total). All measurements on the APB single crystals were performed inside the Linkam stage, purged and filled with dry nitrogen, at room temperature. Halide perovskite compounds are known to degrade upon irradiation with energetic beams[49,50]. We optimized the beamline, spectrometer and measurement settings to obtain at least two iterations of HERFD-XAS spectra, recorded sequentially from the same (initially fresh) spot of a single crystal sample, which do not show observable differences due to beam damage. The measurement duration of each Br $K$ and Pb $L_3$ spectrometer iteration/run was optimized to be ~12 min long. Supplementary Fig. 9 shows Br $K$ and Pb $L_3$ HERFD-XAS spectra, recorded from single crystal MAPB, associated with pristine (iterations 1 and 2) and degraded MAPB (long beam exposure, iteration 13).

**Hard X-ray photoelectron spectroscopy.** Valence band and core level PES measurements were performed at the HIKE endstation attached to the beamline KMC-1 of the synchrotron facility BESSY II[51]. With an excitation energy of 4000 eV, the measurements are expected to be more bulk-sensitive and as-grown (un-cleaved) crystals were used. The overall energy resolution of PES, determined

from the FWHM of Gaussian-broadened Au $4f_{7/2}$ photoelectron lines, is 0.3 eV and is governed primarily by the photon bandwidth of the Si(311) monochromator. Binding energy calibration was performed by recording Au $4f$ spectra from a reference gold foil and setting the binding energy of $E_{Au4f7/2}$ to 84.0 eV. Multiple iterations of core level spectra were recorded, to distinguish the onset of beam damage. See Supplementary Note 7 and Supplementary Fig. 10 and 11 for further details.

**Calculations.** We performed periodic DFT calculations using the Quickstep code in the CP2K package[52–54]. We used the Perdew–Burke–Ernzerhof (PBE) generalized gradient approximation as the exchange-correlation (XC) functional[55]. In addition, we have included a Van der Waals correction using Grimme's D3 method[56]. The geometry optimizations were performed using the Gaussian planewave (GPW) method with Goedecker-Teter-Hutter (GTH) pseudopotentials as well as TZVP-MOLOPT-GTH (for carbon, nitrogen, hydrogen) and DZVP-MOLOPT-GTH (for lead and bromine) basis sets[57–59]. An energy cut-off of 600 Ry was used along with a multi-grid consisting of five grids. The starting geometries for MAPB, FAPB and CsPB are (cubic, $3 \times 3 \times 3$), (cubic, $3 \times 3 \times 3$) and (orthorhombic, $2 \times 2 \times 4$), respectively, taken from the literature[60–62]. All electronic structure calculations were performed at the gamma point of the super cells. Ab initio molecular dynamics simulations were carried out with the same computational parameters within an NPT ensemble at 300 K and 0 atm. The MD simulations were performed for a minimum of 30 ps using a 0.5 fs time step. The drift in the conserved energy is observed to be below $10^{-6}$ a.u. $ps^{-1}$ $atom^{-1}$.

To assess the potential influence of spin-orbit coupling on the calculated electronic structures, total density of states calculations including SOC have been performed using DFT at the GGA level with the Quantum Espresso (QE) package for all three lead bromide perovskites[63]. These calculations employed fully relativistic Rappe–Rabe–Kaxiras–Joannopoulos ultrafast pseudopotentials, including nonlinear core corrections, for all constituent atoms available in the QE pseudopotential library[63,64]. Similar to the CP2K calculations, the PBE exchange-correlation functional and the gamma point of the supercells have been used[55]. We chose plane-wave energy cutoffs of 45 and 455 Ry for the electronic wave function and charge density, respectively.

X-ray absorption spectra were calculated using the Gaussian augmented plane-wave (GAPW) implementation of core-level spectroscopy in CP2K, which allows for a mixed pseudopotential and all-electron description[22,65]. The bromine, hydrogen, carbon and nitrogen atoms were treated at an all-electron level and described with 6-311Gxx basis sets while the GTH pseudopotential with TZVP basis sets were used for the lead and cesium atoms. To simulate core-hole relaxation effects, we use the transition potential method with a half-core hole (TPHH) at the Br $1s$ state. Separate spectral simulations were carried out for each bromine atom in the supercell, where the supercells are obtained from geometry optimization as well as AIMD snapshots (three or four configurations are selected for FAPB, MAPB or CsPB with 10 ps time differences in the AIMD trajectory and close to 100 Br atoms for each configuration). Individual spectra are then averaged together to generate aggregate spectra. The calculated, discrete XAS spectra are convoluted with a Gaussian function with a broadening parameter or σ of 1.0 eV (corresponding to FWHM of 2.355 eV) to simulate experimental spectra. Absorption energies obtained from simulation are underestimated in comparison to experimental data; this is a known limitation of the TPHH approximation. Consequently, an ad-hoc rigid shift of +194.3 eV (~1.5%) was added to all calculated Br $K$ XAS spectra for direct comparison to the experiment.

## Data availability
The processed data presented in this work are available from the corresponding authors on reasonable request. The unprocessed experimental data is associated with the beamtime proposal identifiers listed in the Acknowledgements, and is available from the corresponding authors on reasonable request.

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

## Acknowledgements

The RXES measurements were carried out at beamline P64 of the synchrotron facility PETRA III at DESY, a member of the Helmholtz Association (HGF). The von Hamos-type hard X-ray spectrometer was realized by the group of Prof. Matthias Bauer (University of Paderborn) in the frame of projects FKZ 05K13UK1 and FKZ 05K14PP1, supported by Bundesministerium für Bildung and Forschung. We thank DESY for the provision of beamtime (Proposal No. I-20181028 EC, I-20190356 EC). User (G.J.M., S.M.B., D.P.) experiments at PETRA III have been supported by the project CALIPSOplus under the Grant Agreement 730872 from the EU Framework Programme for Research and Innovation HORIZON 2020. The HAXPES measurements were performed at the HIKE end-station attached to the beamline KMC-1 of the synchrotron facility BESSY II. We thank HZB for the provision of beamtime (Proposal No. 192-08827). S.M.B. and G.J.M. thank the Swedish Research Council (contract 2018-05525) for financial support. H.R., G.J.M., D.P., and S.M. acknowledge the Swedish Research Council (grant # 2018-06465 and # 2018-04330) and the Swedish Energy Agency (P50636) for funding. P.K.N. acknowledges support from the Department of Atomic Energy, Government of India, under Project Identification no. RTI 4007, Science and Engineering Research Board India core research grant (CRG/2020/003877) and Swarna Jayanti Fellowship, DST, India. M.O. acknowledges support from the European Union's Horizon 2020 Research and Innovation programme under the Marie Skłodowska-Curie grant agreement No 860553, and the Swedish Energy Agency (contract 2017-006797). The calculations were enabled by resources provided by the Swedish National Infrastructure for Computing (SNIC) at the Swedish National Supercomputer Center (NSC), the High Performance Computer Center North (HPC2N), and Chalmers Centre for Computational Science and Engineering (C3SE) partially funded by the Swedish Research Council through grant agreement no. 2018-05973. We thank Sigurd Wagner (Princeton) for discussions related to the mechanism of hot carrier cooling in HaPs.

## Author contributions

G.J.M. and S.M.B. conceived and designed the experimental study. P.K.N. grew the single crystals. G.J.M., S.M.B., and A.K. performed the RXES measurements and analyzed the data. D.P., S.M., and G.J.M. performed the HAXPES measurements; D.P. and G.J.M. analyzed the data. J.A. performed the XRD characterization and analysis, under the supervision of P.K.N. C.K. performed AIMD simulations, PDOS and XAS calculations for the snapshots and analyzed all computational results. G.J.M., S.M.B., C.K., and M.O. analyzed the XAS. G.J.M. proposed the links between the findings and questions of technological relevance. G.J.M. and S.M.B. wrote the first draft of the manuscript. All authors contributed to revisions of the manuscript. G.J.M., H.R., M.O., and S.M.B. supervised the project.

## Funding

## Competing interests

The authors declare no competing interests.
