## [Peer Review File · Nature Communications]

Title: A-site cation influence on the conduction band of lead bromide perovskitesREVIEWER COMMENTS

Reviewer #1 (Remarks to the Author):

The paper "A-site Cation Influence on the Conduction Band of Lead Bromide Perovskites" by Man et al. focuses on the problem of the A-cation influence on the electronic structure of three Pb-based perovskites.

The topic of perovskites is of top interest due to their unique solar energy conversion capabilities. However, Pb-based perovskites have been known for many decades and according to my personal knowledge are not real dream systems due to the lead content. The relevance of the subject and the interest to a broad audience, not speaking about cutting edge science or new technologies, might thus be a bit doubted. But I am not an expert in this field and might be wrong here.

Still, while the paper explains some important aspects of selected compounds' electronic structure, it fails to deliver a message of how these results will contribute to the perovskite technology development.

The methodology selection is highly appropriate but also standard and well-established. Moreover, neither the method application nor the analysis provides any novelty to the field. Finally, the manuscript is written in the form of a detailed report and kind of fails to address the general and broad audience of the Nature Communications journal. More likely, after thorough language and argumentation revision, this paper is suitable for a much more specialized journal. Therefore, it is not suitable for publication in Nature Communications in my opinion.

Reviewer #2 (Remarks to the Author):

The authors perform detailed X-ray absorption spectroscopy measurements on lead-bromide-based single crystals, coupled with density functional theory calculations. Through this in-depth analysis, they identify sigma and pi states in the conduction band that shift in relative energy with the size and species of the A-site cation. The authors then extrapolate to relate their findings with observations made in the literature on hot carrier cooling and polaronic transport in these perovskites. This work is certainly interesting and adds to the literature. However, in order to be suitable for Nature Communications, the broader implications of the findings need to be more strongly made.

Major points:

- There have already been several studies looking into the effects of the A-site cation on the structure of these perovskites. E.g., <https://pubs.acs.org/doi/pdf/10.1021/acsmaterialslett.9b00209> and <https://pubs.acs.org/doi/abs/10.1021/acs.chemmater.8b01851> which do not seem to have been cited in the main text. This work is, broadly speaking, along the same lines in focussing on specific materials properties using novel characterisation techniques. In order to be substantially better, the paper will have to demonstrate through direct measurements on the crystal analyzed that there is a correlation between the sigma-pi spacing vs. hot carrier cooling rate or nature of polarons formed (e.g., with

transient absorption spectroscopy measurements)

- The current analysis appears to only consider the static picture. However, MA and FA will rotate/oscillate. The authors should discuss how this will influence the sigma-pi splitting in their crystals and, ultimately, the rate of hot carrier cooling/polaron formation

Minor:

- In the experimental, the atmosphere the crystals were grown in/processed in, and measurements made in need to be given (air, N₂, humidity, etc.)
- Did the authors consider spin orbit coupling in their calculations? If not, how would this influence their results?
- In Figure 2, the calculated and measured XAS spectra do not appear to fit well. Why is that?

Reviewer #3 (Remarks to the Author):

I have read the paper titled "A-site Cation Influence on the Conduction Band of Lead Bromide Perovskites" by Man et al.

The paper focuses, combining a experimental (mainly HERFD-XAS) and theoretical (DFT, AIMD) approach, on a well-debated issue which is the possible role of the A-cation in the optoelectronic features of some lead bromide perovskites (both hybrid and full inorganic).

The paper seems to me quite solid and interesting. The setup is OK. Still I think there is room for improving it. I think that the paper could be potentially accepted once the authors amend it according to some suggestions I provide in the following.

1. Authors start their paper mentioning the hot carrier solar cells and slow hot carrier cooling, a topic previously widely investigated for such class of materials (see Shen et al., Appl. Phys. Lett. 111, 153903 (2017) and Kawai et al., Nano Lett. 15, 3103 (2015)) but then they discuss them in a very short note in Supporting Info. It would be probably useful to expand/comment on possible connections between reported results and possible repercussions in hot carrier solar cell working principles in the main text.

2. Related to the previous point, the feature of slow hot-carrier cooling has been previously debated in similar (mainly iodine based perovskites) systems and in such papers both experimentally and theoretically (those mentioned above and that reported in ref 21 of SI) hot-carrier lifetimes are found to highly correlate with DOS of the systems, i.e. the lifetime increases as the DOS becomes smaller at the band edge.

In the present paper, to strengthen their message, authors report both in Fig 3 and SF3 PDOS of Occupied/Unoccupied states for the strong/weak H-bonds Br/A-cation for the three halide perovskites, and this is pretty fine to me.

In my opinion their analysis would be definitely more complete just adding the total calculated DOS profile of the three systems. In this way the reader could have a better overview, understanding PDOS positioning in the total DOS plot and additionally could have a qualitative/quantitative idea of the lifetime of

carriers in such systems (function, as stated, of the VBM/CBM DOS shape). This point would enhance the quality of their manuscript, extending the interesting discussion in SN6, where the connection between σ - π characters of the bandgap and the mechanism of slow hot electron cooling is discussed

3. I have some concerns about the point that should be somehow the main message of the paper. In SN2, page 4, the connection between experiments and theory seems to me a bit weak. First because the peak position changes are really small (0.1 eV) and also because they do not reproduce, if not as trend, the experimental data. I would say that the conclusion authors get is not so strong (at least if based on the comparison between theory and experiments). Probably here authors should have to be more convincing

4. I find the theoretical setup quite reasonable. I have anyway a couple of questions I would like the authors to answer.

To which extent does DFT underestimate excited state properties here? I mean, Fig 3 shows unoccupied state calculated PDOS. And as we know DFT underestimates the description of the conduction region. So I wonder if authors may discuss the impact of DFT shortcomings on their results. Similarly, what about the lack of relativistic effects in the calculations? As far as I understand (I could be wrong) authors do not include SOC effects in the electronic features of their systems. I feel that, considering the giant spin-orbit coupling (SOC) in the conduction-band reported in such systems, also the interactions between A-cation and the Pb-Br network should result highly impacted (I think that SOC should scroll down the heavy atom levels, leaving almost unaltered the A-site cation orbital positioning, thus disentangling the two contributions).

Authors use AIMD and extract trajectories and on different configurations they average to calculate the Br K XAS spectra. Exactly, how many configurations are used to do it? Is there any chance that doing this calculation only on a reduced number of configurations the spectra may result underestimated?

Furthermore, I think that a timestep of 0.5 fs is still too large in AIMD simulations: this because CH₃ vibrations in methylammonium should not be captured with a similar time step. I think that 0.125 or 0.25 fs would have worked much better. I would like the authors to comment about this latter aspect, too

5. Last but not least (but minor). In Fig. 2f, I can understand the σ -symmetry states (where they are located). At the same time, I am not able to see where the π ones are located in the big picture: the inset in my opinion is too small, I would enlarge it or otherwise find a way to better show the π state positioning in the structure. To which Br/A-cation atoms in the main picture do they refer to?

Text inserted into the manuscript and Supplementary Information, to address reviewer
questions, has been highlighted yellow, green and blue for Reviewers #1, #2 and #3. Minor
additions and reformulations which enhance the clarity of the manuscript are highlighted in gray.

**[1] Reviewer #1:** The paper "A-site Cation Influence on the Conduction Band of Lead Bromide
Perovskites" by Man et al. focuses on the problem of the A-cation influence on the electronic
structure of three Pb-based perovskites.

The topic of perovskites is of top interest due to their unique solar energy conversion
capabilities. However, Pb-based perovskites have been known for many decades and according
to my personal knowledge are not real dream systems due to the lead content. The relevance of
the subject and the interest to a broad audience, not speaking about cutting edge science or
new technologies, might thus be a bit doubted. But I am not an expert in this field and might be
wrong here.

Still, while the paper explains some important aspects of selected compounds' electronic
structure, it fails to deliver a message of how these results will contribute to the perovskite
technology development.

The methodology selection is highly appropriate but also standard and well-established.
Moreover, neither the method application nor the analysis provides any novelty to the field.
Finally, the manuscript is written in the form of a detailed report and kind of fails to address the
general and broad audience of the Nature Communications journal. More likely, after thorough
language and argumentation revision, this paper is suitable for a much more specialized journal.
Therefore, it is not suitable for publication in Nature Communications in my opinion.

**[1] Response:** We thank Reviewer #1 for their comments and for their appreciation of the
appropriateness of our methodology. As Reviewer #1 has noted, Pb-based perovskites have been
known for many decades (D. Weber, Z. Naturforsch. **33b**, 1443 (1978)), though many fundamental
questions and hence opportunities to perform cutting edge science remain (D.A. Egger *et al.*, Adv.
Mater. **30**, 1800691 (2018)). Our work provides new insights into two fundamental questions: (i) what is
the optoelectronic role of the A-cation and (ii) what is the mechanism of slow hot carrier cooling in
halide perovskites (HaPs)? Since 2015/16, multiple A-cation HaP films have been used in devices (Saliba
*et al.*, Energy Environ. Sci. **9**, 1989 (2016)), primarily to enhance their ambient stabilities, and the
optoelectronic role of the A-cation, a well-debated issue, remains controversial. Our work provides
conclusive evidence of A-cation effects on the conduction band states, relevant for electron dynamics.
Slow hot carrier cooling in halide perovskites (HaPs) has been reported nearly a decade ago (Xing *et al.*,
Science **342**, 344 (2013)) and HaPs show hot carrier decay constants orders of magnitude higher than all
competing materials used in solar cells. The mechanism of slow hot carrier cooling in HaPs remains
controversial. Hot carrier solar cells, which require the existence of slow hot carrier cooling in the solar
absorber, have potential to surpass the thermodynamic power conversion efficiency limit of ~33%
(Shockley-Queisser limit) and were proposed 40 years ago (Ross, Nozik, J. Appl. Phys. **53**, 3813 (1982)).
To-date, no actual devices exist (Ferry *et al.*, J. Appl. Phys. **128**, 220903 (2020)). Our work sheds new
light on the mechanism of slow hot carrier cooling in HaPs, which are the most promising class of
materials for hot carrier photovoltaics to-date. While the prototypical HaPs are lead-based, perhaps the
solar cells can be encapsulated well, if the technological advantages outweigh the costs. Cadmium
telluride solar panels from First Solar, which contain toxic cadmium, are deployed in utility-scale solar
power plants with expected plant lifetimes of > 20 years. More importantly, our findings apply to the

entire class of halide perovskites and will hopefully stimulate further work into halide perovskite
subfamilies containing other B-site atoms (e.g. tin). We have created new text which delivers a message
of how our results contribute to perovskite technology development.

As for novelty, our findings exist solely due to our use of high quality HaP single crystals and advanced,
synchrotron-based high-resolution X-ray absorption spectroscopy (HERFD-XAS). High quality crystals are
difficult to obtain and are crucial for obtaining reliable measurements as this class of materials is
susceptible to beam damage (Klein-Kedem *et al.*, *Acc. Chem. Res.* 49, 347 (2016)). The measurements
necessitate the use of a synchrotron-based beamline and endstation, which are less common than
home-lab characterization techniques. We know only of two experimental techniques which can probe
the conduction band states: XAS and inverse photoelectron spectroscopy (IPES), where XAS offers
elemental- and orbital-selectivity. Inverse PES has been applied (Endres *et al.*, *JPCL* 7, 2722 (2016)) but
the optoelectronic role of the A-cation remained unclear, hence we have resorted to the use of XAS.

We have revised the text, language and argumentation, aided by questions from Reviewers #2 and #3.
We have further strengthened our work for a broad audience, which includes (i) physical science
communities engaged in research with advanced spectroscopy, (ii) chemical/materials science
communities engaged primarily with materials research, (iii) participators/spectators in both
fundamental and applied halide/non-halide perovskite research (spanning materials synthesis to device-
relevant interface engineering to advanced materials characterization), (iv) participators/spectators in
photovoltaic and general optoelectronic materials research, and more. Hence we believe our work is
suitable for publication in Nature Communications instead of a more specialized journal.

66 // Modified text in the Introduction

*We discuss the connection between σ - π splitting and slow cooling of hot electrons, which is relevant for*
*potential HaP-based hot carrier solar cells which could surpass the Shockley-Queisser thermodynamic*
*limit.*

71 // New text inserted into Discussion

*Single-junction solar cells, irrespective of device architecture and solar absorber material, are*
*thermodynamically limited in power conversion efficiency (PCE) to slightly above 30% (Shockley and*
*Queisser, *J. Appl. Phys.* 32, 510 (1961)). This limit assumes all excess energy from above-bandgap light*
*absorption is unavailable to do work. Current certified PCE for thin-film crystal GaAs, single-crystal Si,*
*and halide perovskite solar cells are 29.1 %, 26.7 % and 25.7 %, respectively (NREL). While the use of*
*terrestrial-based, non-concentrator triple-junction solar cells (39.5 % PCE) is possible, and*
*silicon/perovskite tandem solar cells (29.8% PCE) are being commercialized, single-junction cells with PCE*
*surpassing the Shockley-Queisser limit are highly attractive for techno-economic reasons. If the energy*
*from above-bandgap photoexcitation is available to do work, the thermodynamic limit could be*
*substantially higher (Ross and Nozik, *J. Appl. Phys.* 53, 3813 (1982)). Actual hot carrier solar cells have*
*yet to be demonstrated, though the concept was proposed forty years ago (Ferry *et al.*, *J. Appl. Phys.**
*128, 220903 (2020)).*

*The development of a hot carrier solar cell hinges on several prerequisites, such as the existence of slow*
*cooling of hot carriers in the light absorber and efficient extraction of hot carriers at device interfaces (Li*
*et al., Adv. Mat. 31, 1802486 (2019)). At comparable electron temperatures of ~1000 K, hot carrier*
*cooling time constants are considerably longer for HaPs (100's of ps) versus state-of-the-art GaAs (< 10*
*ps) (Zhu et al., Science 353, 1409 (2016), Rosenwaks et al., PRB 48, 14675 (1993)). This suggests HaP-*
*based solar cell technology has the potential to become the leading single-junction photovoltaic*
*technology. Slow hot carrier cooling in HaPs has been experimentally observed, and phenomenologically*
*explained with the polaronic screening and hot phonon bottleneck mechanisms (Xing et al., Science 342,*
*344 (2013); Zhu et al., Science 353, 1409 (2016); Yang et al., Nat. Photonics 10, 53 (2016)). Our finding,*
*of the σ - π splitting in the conduction band, provides an alternative mechanism for electron cooling, thus*
*advancing the general understanding of electron dynamics in HaPs and spurring technological*
*development of HaP-based hot carrier solar cells. Simultaneously, our study highlights the mostly-*
*overlooked optoelectronic role of the A-cation, shedding light on a well-debated issue and providing*
*guidance to HaP application developers who are tailoring HaP properties for various optoelectronic*
*applications.*

**[2] Reviewer #2:** The authors perform detailed X-ray absorption spectroscopy measurements
on lead-bromide-based single crystals, coupled with density functional theory calculations.
Through this in-depth analysis, they identify sigma and pi states in the conduction band that shift
in relative energy with the size and species of the A-site cation. The authors then extrapolate to
relate their findings with observations made in the literature on hot carrier cooling and polaronic
transport in these perovskites. This work is certainly interesting and adds to the literature.
However, in order to be suitable for Nature Communications, the broader implications of the
findings need to be more strongly made.

**[2] Response:** We thank Reviewer #2 for their effort in reviewing our paper, and for their interest in our
work. We agree that the impact of our results should be stated more clearly and have elaborated on the
broader implications of the findings in points [3] and [4].

We have further strengthened our work for a broad audience: ranging from communities engaged in
research using advanced spectroscopy to communities engaged primarily with materials research,
participators/spectators in both fundamental and applied halide and non-halide perovskite research,
and participators/spectators in photovoltaic and general optoelectronic materials research. Hence we
believe our work is suitable for publication in Nature Communications. See also our response to
Reviewer #1.

**[3] Reviewer #2:** Major points:

• There have already been several studies looking into the effects of the A-site cation on the
structure of these perovskites. E.g.,
<https://pubs.acs.org/doi/pdf/10.1021/acsmaterialslett.9b00209> and
<https://pubs.acs.org/doi/abs/10.1021/acs.chemmater.8b01851> which do not seem to have been
cited in the main text. This work is, broadly speaking, along the same lines in focussing on

specific materials properties using novel characterisation techniques. In order to be substantially
better, the paper will have to demonstrate through direct measurements on the crystals
analyzed that there is a correlation between the sigma-pi spacing vs. hot carrier cooling rate or
nature of polarons formed (e.g., with transient absorption spectroscopy measurements)

**[3] Response:** We thank Reviewer #2 for bringing the two works to our attention. We have cited them
in the new text below. As for the correlation between the sigma-pi splitting versus hot carrier cooling
rate, we find strong experimental support in the literature. We have added the following new text.

134 // New text inserted into the results section titled "Results of overall A-cation influence on the crystal
and electronic structures"

*A-cation influence on the crystal structure and optoelectronic functionality has previously been*
*investigated (Ghosh et al., Chem. Mater. 30, 5194 (2018); Mozur et al., ACS Mat. Lett. 2, 260 (2019)).*
*Our XRD, Br K and Pb L3 HERFD-XAS, and Br Kβ1 measurements show that symmetry lowering of the Br-*
*Pb framework increases Br-Pb covalency and shifts the CBM up in potential energy, which is consistent*
*with the findings of a pure computational study (Ghosh et al.). Hence, we find A-cation effects on the*
*crystal and electronic structures that are generally consistent with previous findings. Additionally, we*
*provide direct measurements of the Br-Pb bond ionicity/covalency with XES and Br K-edge HERFD-XAS*
*measurements of the bromine-projected conduction band states.*

As for the correlation between the sigma-pi splitting versus hot carrier cooling rate, we find strong
experimental support in the literature. We have added the following new text.

147 // New text inserted into Discussion

[revised manuscript text omitted]

**[4] Reviewer #2:** • The current analysis appears to only consider the static picture. However,
MA and FA will rotate/oscillate. The authors should discuss how this will influence the sigma-pi
splitting in their crystals and, ultimately, the rate of hot carrier cooling/polaron formation

**[4] Response:** We interpret static to mean time-averaged (as both our measurements and calculations,
sampled over ab initio MD simulations, probe time-averaged electronic structure). We interpret
Reviewer #2's comment to mean "how picosecond MA/FA rotations/oscillations will influence the time-
averaged σ - π splitting, and ultimately the rate of picosecond hot carrier cooling, polaron formation?"

To incorporate the effects of picosecond MA/FA rotation/oscillation, our DFT-based *ab initio* molecular
dynamics (AIMD) simulations, comprised of large structural models (supercells) containing multiple
MA/FA molecules, were performed at room temperature for long simulation times (minimum of 30 ps).
Further details can be found in the Methods section. The spectral calculations were averaged over three
or four snapshots of supercells, hence the effect of the molecular rotations/vibrations on the spectra
and electronic structure are included in a time-averaged manner.

Since the σ - π splitting presented here, experimentally and computationally, is a time-
averaged/persistent feature in the electronic structure, we expect all dynamical processes involving

above-band-edge electrons (e.g. hot electron cooling, possible polaron formation, etc.) to be influenced
by the σ - π splitting, irrespective of their timescale(s).

We have added the following text into the main body.

210 // New text inserted into the section titled "Calculated ground-state electronic structure"

*We find a positive correlation between the average strength of N-H ... Br hydrogen bonding and the*
*average magnitude of the σ - π splitting and deduce that the rotation/oscillation of the organic A-cations*
*likely do not have a causal relationship with the magnitude of the splitting. Further details can be found*
*in Supplementary Discussion 1, where we also argue that the σ - π splitting is a time-averaged feature in*
*the conduction band states and expect it to influence all electron dynamics irrespective of timescale.*

217 // New text inserted into "Supplementary Discussion 1. Potential influence of organic A-cation rotations
and oscillations on the σ - π splitting and hot carrier cooling rate."

*The timescales of organic cation rotation/oscillation in MAPB and FAPB have been found, via IR*
*spectroscopy, solid-state NMR, etc. to be in the range of 0.3 – 2 ps and 0.1 – 2 ps, respectively (Gallop et*
*al., JPCL 9, 5987 (2018)). Our HERFD-XAS measurements were recorded with a 1 second*
*integration/accumulation time per energy point, whereas the XAS process is ultrafast. Likewise, the*
*density of states and XA spectrum simulations are based on instantaneous snapshots from ab initio*
*molecular dynamics simulations. Consequently, we are sampling the time-averaged electronic structure*
*over virtually instantaneous configurations, and find the σ - π splitting to be a time-averaged feature in*
*the conduction band. This is analogous to sampling the time-averaged crystal structure of HaPs with*
*bulk X-ray diffraction, which shows highly crystalline long-range order in spite of the picosecond-*
*timescale structural disorder (Yaffe et al., PRL 118, 136001 (2017), Singh et al., PRB 101, 054302 (2020)).*

*In the theoretical simulations, we find a positive correlation between the strength of N-H ... Br hydrogen*
*bonding and the time-averaged magnitude of the σ - π splitting: CsPB (no H-bonding, σ - π _{calculated} = 3.5 eV)*
*→ MAPB (H-Br bond distance ~2.47 Å, σ - π _{calculated} = 4.0 eV) → FAPB (H-Br bond distance ~ 2.37 Å, σ -*
*π _{calculated} = 4.2 eV). Thus, we suggest that the average strength of hydrogen-bonding strongly influences*
*the average magnitude of the σ - π splitting. Organic A-cation rotation/oscillation rates may not strongly*
*influence the magnitude of σ - π splitting. Methylammonium and formamidinium have comparable*
*rotation/oscillation time constants (Gallop et al.) and ionic radii ($r_{MA^+} = 2.70$ Å, $r_{FA^+} = 2.79$ Å), but FA⁺*
*hydrogen-bonds more strongly to the halide framework (Amat et al., Nano Lett. 14, 3608 (2014)). Since*
*the σ - π splitting is a persistent/time-averaged feature in the conduction band, it is expected to influence*
*all electron dynamics (e.g. hot electron cooling rate, potential polaron formation), irrespective of their*
*timescale(s).*

**[5] Reviewer #2:** Minor:
• In the experimental, the atmosphere the crystals were grown in/processed in, and
measurements made in need to be given (air, N₂, humidity, etc.)

**[5] Response:** We thank Reviewer #2 for their attention to detail. We have added the following text to
the Methods.

248 // Modified text in "Methods (APB crystal growth)"

*The MAPB and FAPB single crystals were solution-grown using methods reported by Dr. Pabitra Nayak*
*previously (Nayak et al., Nat. Commun. 7, 13303 (2016); Kabakova et al., J. Mater. Chem. C 6, 3861*
*(2018)). The CsPB single crystals were solution-grown using a method reported by others (Dirin et al.,*
*Chem. Mater. 28, 8470 (2016)). All single crystals were grown in an ambient air environment, then*
*transferred into vials of chlorobenzene for storage and preservation.*

255 // New text inserted into "Methods (Hard x-ray absorption and emission spectroscopy)"

*Single crystals were prepared for measurements by first extracting them from chlorobenzene, blowdrying*
*with compressed air/nitrogen in an ambient air environment, then mounting them into a Linkam T95*
*heating/cooling stage (~ 5 minutes total). All measurements on the APB single crystals were performed*
*inside the Linkam stage, purged and filled with dry nitrogen, at room temperature.*

**[6] Reviewer #2:** • Did the authors consider spin orbit coupling in their calculations? If not, how
would this influence their results?

**[6] Response:** We thank Reviewer #2 for raising this important point. We had not incorporated spin-
orbit-coupling (SOC) effects into the calculations presented in the previously submitted version.

Now we have calculated the total electronic density of states with and without spin-orbit coupling (SOC)
for all three lead bromide perovskites using DFT at the GGA level (using the Quantum Espresso package);
the results are shown below and in Supplementary Fig. 5. We find that the influence of SOC is almost
negligible in the conduction band region and is very small in the valence band region (0 to -10 eV).
However, we clearly observe the expected splitting of the Pb 5*d* shallow core level in the range of -12 to
271 -20 eV. Thus, we conclude that all of the discussions related to the electronic structure and X-ray
absorption spectra for these three materials as well as the conclusions drawn in the previously
submitted version remain unaffected.

As for the crystal structures, geometry optimization of MAPI (related compound) via DFT with and
without SOC effects results in similar Pb-I bond distances (Umari et al., Sci. Reports 4, 4467 (2015)).
Hence, we expect the crystal structures to be modeled reasonably well by DFT without SOC. Other
work, which computationally demonstrate the presence of bond disproportionation in lead halide
perovskites with DFT, did not incorporate SOC effects as well (Dalpian et al., PRB 98, 075135 (2018)).

Consequently, we expect our DFT-based molecular dynamics simulations of the crystal structures to be
unaffected by the absence of SOC effects.

In the revised manuscript, we have added the following statements and details of calculations.

283 // New text inserted into the section titled "Calculated ground-state electronic structure"

*To evaluate the possible influence of spin-orbit coupling (SOC) on the conclusions of our analysis, we*
*performed total electronic density of states (TDOS) calculations for all three lead bromide perovskites*
*using DFT with/without SOC effects. Further details are found in the Methods. The results of the*
*calculations, shown in Supplementary Figure 5, suggest that the influence of SOC is negligible for the*
*conduction band region, and hence for the analysis of the Br K-edge XAS data, and it is very small for the*
*valence band region up to -10 eV. The computed TDOS shows the Pb 5d splitting around -12 to -20 eV,*
*which is consistent with the shallow core photoelectron spectra of Pb 5d (Supplementary Figure 6) and*
*confirms the inclusion of SOC in the TDOS calculations.*

293 // New text inserted into "Methods (Calculations)"

*To assess the potential influence of spin-orbit coupling on the calculated electronic structures, total*
*density of states calculations including SOC have been performed using DFT at the GGA level with the*
*Quantum Espresso (QE) package for all three lead bromide perovskites (Giannozzi et al., J. Phys.*
*Condens. Matter 29, 465901 (2017)). These calculations employed fully relativistic Rappe-Rabe-Kaxiras-*
*Joannopoulos ultrafast pseudopotentials, including nonlinear core corrections, for all constituent atoms*
*available in the QE pseudopotential library (Giannozzi et al., J. Phys. Condens. Matter 29, 465901 (2017),*
*Rappe et al., PRB 41, 1227 (1990)). Similar to the CP2K calculations, the PBE exchange-correlation*
*functional (Perdew et al., PRL 77, 3865 (1996))) and the gamma point of the supercells have been used.*
*We chose plane-wave energy cutoffs of 45 and 455 Ry for the electronic wave function and charge*
*density, respectively.*

**Supplementary Fig. 5. Calculated total electronic density of states of CsPB, MAPB, FAPB with and**
 **without spin-orbit coupling effects.** Occupied states are displayed in the left panel, are enumerated
 with negative binding energies and are Gaussian-broadened by 0.3 eV. Unoccupied states are displayed
 in the right panel and are Gaussian-broadened by 1.0 eV to emphasize the main features of interest.
 Calculated total density of states (TDOS) without spin-orbit coupling (SOC) effects are shown with
 dashed lines, and TDOS with SOC effects are shown with solid lines. The calculations were performed
 with Quantum Espresso.

**[7] Reviewer #2:** • In Figure 2, the calculated and measured XAS spectra do not appear to fit
 well. Why is that?

**[7] Response:** In all cases, Figure 2c-e, we observe that the heights of the main-edge profile (up to
 ~ 13476 eV) relative to the post-edge region (above ~ 13476 eV) appear smaller in the calculated spectra
 than in experiment. Similar shortcomings in modeling the post-edge region using the transition
 potential approximation have been reported for nitrogen *K*-edge XAS of MAPbBr₃ and the related
 compound MAPbI₃ (Sterling *et al.*, JPCS 125, 8360 (2021) and references therein); these originate from
 difficulties to efficiently yet accurately model the core-excited states, in particular their relaxation, in
 condensed phases. All modern approaches to calculate core-hole potentials and their related effects
 have both advantages and limitations (Shirley *et al.*, Int. Tables Crystallogr. I (2021)).

Since we are focused on relative comparisons of the main-edge features in a set of materials where the
 A-cation is systematically varied, and primarily discuss the pre- and main-edge features, we do not
 expect the limitations in the computational framework to strongly influence the conclusions.

The aim of electronic structure calculations and X-ray absorption spectra simulations is to identify the
character of the important molecular orbitals involved in the transitions underlying to the experimental
spectra. We use the density functional theory at GGA level along with the half core-hole transition
potential (TPHH) method. This approach gives quite satisfactory spectra for several systems including
solids. It also allows us to perform calculations for reasonably large periodic systems like the structural
models (supercells considered in the present study) for which computational challenges of higher-level
quantum chemical methods may pose a severe limitation. Most of the important discussions and
conclusions drawn in the manuscript involves electronic structure of the main peaks. It is a well-known
problem/limitation that the electronic states in the conduction bands (corresponding to the post-edge
regions) which are far away from the Fermi level are not well reproduced by DFT due to approximate
exchange-correlation (GGA) functional as well as usage of finite basis sets which may not give good
description for continuum states. Also the variation in electronic relaxation in core-excitations of
different character is challenging to capture. These factors lead to mismatch in the post-edge region. It
is also noted that the experimental spectra are also nearly featureless in the post-edge region.

We have revised Supplementary Note 2 (text copied-and-pasted below) to address the match between
calculated and measured XA spectra.

355 // Revised "Supplementary Note 2: Comparison between experimental and calculated Br K XAS spectra
of FAPB, MAPB and CsPB."

*In all cases, Fig. 2c-e, we observe that the heights of the main-edge profile (up to ~13476 eV) relative to*
*the post-edge region (above ~13476 eV) appear smaller in the calculated spectra than in experiment.*
*We also see in the calculations that the post-edge in each system, above ~13476 eV, is built up from two*
*bands with σ -symmetry and a central π -symmetry band, but that is not resolved in the experiment. The*
*small π -bonding contributions at similar excitation energies to the σ -bonding contributions in the main-*
*edge, from ~13464 to ~13469 eV, may appear because the Pb-Br-Pb is bent.*

*We examine the computed peak positions of the σ main-edge feature, the main constituent of the*
*absorption onset, and find a trend where the peak position increases, going from FAPB (13468.2 eV) to*
*MAPB (13468.3 eV) to CsPB (13468.4 eV). The A-cation-influenced trend in the σ - π splitting, one of the*
*key findings in our work, is unaffected by the Br K absorption onset (where our calculations show 0.1 eV*
*relative changes) as the splitting spans several eV of the conduction band region. Our XAS calculations*
*do correctly reproduce the absorption onset trend (increases in this order: CsPB \rightarrow MAPB \rightarrow FAPB).*
*While the relative differences between MAPB and FAPB (experiment: 0.5 eV, calculation: 0.1 eV) were*
*not fully captured, the relative differences between CsPB and MAPB (experiment: 0.1 eV, calculation: 0.1*
*eV) were fully captured. Overall, the quantitative σ - π splitting trend and the qualitative absorption onset*
*trend hold.*

*The transition potential DFT methodology has been demonstrated to give useful support to the analysis*
*of experimental data in numerous applications: (i) different element K-edge XAS and (ii) molecular and*
*condensed phases. It is an approximate method, but has the advantage that it often gives semi-*
*quantitative results combined with an excellent scaling to large systems. Ekimova et al. (JACS 139,*
*12773 (2017)) used a similar methodology to investigate hydrogen-bonded solutes in an environment of*
*63 water molecules, and Hou et al. (PRB 87, 165401 (2013)) used a similar methodology to investigate*

*nitrogen-doped graphene. Dalpian et al. (PRB 98, 075135 (2018)) have shown that it is essential to use a*
*large supercell for accurately modeling the ground-state crystal structures of HaPs; this necessitates the*
*use of methods with excellent scaling characteristics.*

**[8] Reviewer #3:** I have read the paper titled "A-site Cation Influence on the Conduction Band
of Lead Bromide Perovskites" by Man et al.

The paper focuses, combining a experimental (mainly HERFD-XAS) and theoretical (DFT,
AIMD) approach, on a well-debated issue which is the possible role of the A-cation in the
optoelectronic features of some lead bromide perovskites (both hybrid and full inorganic).

The paper seems to me quite solid and interesting. The setup is OK. Still I think there is room
for improving it. I think that the paper could be potentially accepted once the authors amend it
according to some suggestions I provide in the following.

**[8] Response:** We thank Reviewer #3 for their effort in reviewing our work and for their interest. We
have addressed the amendments in responses [9] to [16].

**[9] Reviewer #3:** 1. Authors start their paper mentioning the hot carrier solar cells and slow hot
carrier cooling, a topic previously widely investigated for such class of materials (see Shen et
al., Appl. Phys. Lett. 111, 153903 (2017) and Kawai et al.,
Nano Lett. 15, 3103 (2015)) but then they discuss them in a very short note in Supporting Info. It
would be probably useful to expand/comment on possible connections between reported results
and possible repercussions in hot carrier solar cell working principles in the main text.

**[9] Response:** We thank Reviewer #3 for bringing the two works to our attention. We have added the
following text into the main body, including text that connects the aforementioned works to our work.
We have also discussed the possible repercussions of our findings on practical HaP-based hot carrier
solar cell design.

**// New text inserted into "Discussion"**

[revised manuscript text omitted]

**[10] Reviewer #3:** 2. Related to the previous point, the feature of slow hot-carrier cooling has
been previously debated in similar (mainly iodine based perovskites) systems and in such
papers both experimentally and theoretically (those mentioned above and that reported in ref 21
of SI) hot-carrier lifetimes are found to highly correlate with DOS of the systems, i.e. the lifetime
increases as the DOS becomes smaller at the bandedge.

In the present paper, to strengthen their message, authors report both in Fig 3 and SF3 PDOS
of Occupied/Unoccupied states for the strong/weak H-bonds Br/A-cation for the three halide
perovskites, and this is pretty fine to me.

In my opinion their analysis would be definitely more complete just adding the total calculated
DOS profile of the three systems. In this way the reader could have a better overview,
understanding PDOS positioning in the total DOS plot and additionally could have a quali-
489 /quantitative idea of the lifetime of carriers in such systems (function, as stated, of the VBM/CBM
DOS shape). This point would enhance the quality of their manuscript, extending the interesting
discussion in SN6, where the connection between σ - π characters of the bandgap and the
mechanism of slow hot electron cooling is discussed

**[10] Response:** We thank Reviewer #3 for their appreciation of the novelty of the discussion in
Supplementary Note 6. As per Reviewer #3's suggestion, we have created a plot (shown below)
containing both the total density of states and the partial density of states of the three lead bromide
perovskites.

**// New text added into the section titled “Calculated ground-state electronic structure”**

*The Br 4p PDOS comprises the bulk of the valence band states, as shown in Supplementary Figure 4,*
*which displays the Br 4p PDOS and the total DOS (contributions from all elements).*

Supplementary Fig. 4. **Calculated ground-state bromine 4s, 4p σ and 4p π projected density of states**
 **and total density of states of CsPB, FAPB and MAPB.** Occupied states are Gaussian-broadened by 0.3
 506 eV. The top of the valence band is aligned to 0 eV binding energy. Projected density of states (PDOS)
 corresponding to strongly hydrogen-bonded bromine with organic molecules are shown. Total density
 of states (TDOS) are shown with symbols while the PDOS are shown with solid and dashed lines. The
 MAPB and FAPB TDOS features between -3.5 to -6 eV originate from the organic A-cations. The
 calculations were performed with CP2K.

**[11] Reviewer #3:** 3. I have some concerns about the point that should be somehow the main
 message of the paper. In SN2, page 4, the connection between experiments and theory seems
 to me a bit weak. First because the peak position changes are really small (0.1 eV) and also
 because they do not reproduce, if not as trend, the experimental data. I would say that the
 conclusion authors get is not so strong (at least if based on the the comparison between theory
 and experiments). Probably here authors should have to be more convincing

**[11] Response:** We appreciate Reviewer #3's concern, and have revised and added to the text in
 Supplementary Note 2 to address the concerns. Please refer to response [7] for further details.

**[12] Reviewer #3:** 4. I find the theoretical setup quite reasonable. I have anyway a couple of
questions I would like the authors to answer.

To which extent does DFT underestimate excited state properties here? I mean, Fig 3 shows
unoccupied state calculated PDOS. And as we know DFT underestimates the description of the
conduction region. So I wonder if authors may discuss the impact of DFT shortcomings on their
results.

**[12] Response:** We appreciate Reviewer #3's concerns about DFT shortcomings, and have created a
new Supplementary Note to address the concerns.

532 // New text inserted into the results section titled "Calculated ground-state electronic structure"

*In Supplementary Note 3, we briefly discuss potential limitations to DFT calculations of the conduction*
*band electronic structure, and deduce that the σ - π splitting trend is unaffected due to its magnitude of*
*several eV.*

537 // New text inserted into "Supplementary Note 3: Potential DFT-related limitations with the calculation
of the conduction band electronic structure."

*The common exchange-correlation functionals used in DFT are known to yield underestimated band-*
*gaps. However, the occupied and unoccupied parts of the computed DOS are generally in good*
*agreement with experimental photoelectron spectra, stretched by a few percent (Endres et al., JPCL 7,*
*2722 (2016)). Within the conduction band region, systematic DFT underestimation of a few percent or ~*
*0.1 eV is not expected to affect the σ - π splitting trend as the magnitude of the splitting is ~ 4 eV.*

*DFT, using the same exchange-correlation functional as what we have used (Perdew-Burke-Ernzerhof*
*(PBE) generalized gradient approximation (GGA)), has been demonstrated to be reasonably accurate at*
*modeling the conduction band electronic structure of MAPI, a closely related compound (Yang et al., JPCL*
*12, 3773 (2021)). While there is overall agreement between the calculated conduction band dispersions*
*and the measured bands (obtained via angle-resolved inverse photoelectron spectroscopy) over a ~3 eV*
*region of the conduction band, the agreement is not perfect at the 0.1 eV order-of-magnitude energy*
*scale.*

**[13] Reviewer #3:** Similarly, what about the lack of relativistic effects in the calculations? As far
as I understand (I could be wrong) authors do not include SOC effects in the electronic features
of their systems. I feel that, considering the giant spin-orbit coupling (SOC) in the conduction-
band reported in such systems, also the interactions between A-cation and the Pb-Br network
should result highly impacted (I think that SOC should scroll down the heavy atom levels,
leaving almost unaltered the A-site cation orbital positioning, thus disentangling the two
contributions).

**[13] Response:** We thank Reviewer #3 for raising this important point. We had not incorporated spin-
orbit-coupling (SOC) effects into the calculations presented in the previously submitted version. Now
we have calculated the total electronic density of states with and without spin-orbit coupling (SOC) for
all three lead bromide perovskites using DFT at the GGA level (using the Quantum Espresso package).
We find that the influence of SOC is almost negligible in the conduction band region and is very small in
the valence band region (0 to -10 eV). However, we clearly observe the expected splitting of the Pb 5d
shallow core level in the range of -12 to -20 eV. Thus, we conclude that all of the discussions related to
the electronic structure and X-ray absorption spectra for these three materials as well as the conclusions
drawn in the previously submitted version remain unaffected. Please refer to response [6] for further
details.

**[14] Reviewer #3:** Authors use AIMD and extract trajectories and on different configurations
they average to calculate the Br K XAS spectra. Exactly, how many configurations are used to
do it? Is there any chance that doing this calculation only on a reduced number of configurations
the spectra may result underestimated?

**[14] Response:** In our work, three or four configurations were selected for each compound, with 10 ps
time differences in the AIMD trajectory. Since each configuration implies a sampling over Br K-edge XAS
for 81 to 96 bromine atoms, a large number of local Bromide environments are explored, and hence
even a few snapshots are sufficient for obtain a representative average. We have added the following
text to provide additional information to the reader.

581 // New text inserted into the results section titled "Calculated X-ray absorption spectra"

*To capture the structural dynamics of the A-cation accurately, we use three or four configurations, each*
*involving between 81 to 96 Br atoms, with 10 ps time differences in the AIMD trajectory. The longest*
*time constant of organic cation rotation/oscillation in MAPB and FAPB has been found, via IR*
*spectroscopy, solid-state NMR, etc. to be ~ 2 ps (Gallop et al., JPCL 9, 5987 (2018)). The 10 ps time*
*difference covers multiple distinct rotations/oscillations, enabling us to capture the structural dynamics*
*in a time-averaged manner.*

**[15] Reviewer #3:** Furthermore, I think that a timestep of 0.5 fs is still too large in AIMD
simulations: this because CH₃ vibrations in methylammonium should not be captured with a
similar time step. I think that 0.125 or 0.25 fs would have worked much better. I would like the
authors to comment about this latter aspect, too

**[15] Response:** We appreciate Reviewer #3's concern. Our main aim of performing AIMD simulation is to
generate several possible geometrical configurations for organic-inorganic lead bromide perovskites at
room temperature. So that this will capture and include the vibrational and rotational motion, especially
of organic cations. Later, simulate X-ray absorption spectra for these configurations and average them to

make better comparison with experimental spectra measured at the room temperature. We did not
intend to calculate any dynamical properties where the shorter time step is crucial for properly integrating
the equation of motion. We wish to mention here that we have used the SCF convergence threshold of
10^{-6} and the time step of 0.5 fs which is standard (VandeVondele *et al.*, J. Chem. Phys. 122, 014515
(2005)). In order to verify the correctness of the 0.5 fs time step used, we have investigated the drift in
the conserved energy in the simulations with time. The drift in the total energy of the system can be
directly correlated to the issues/errors in the integration. The drift was observed to be below 1.02×10^{-6}
a.u. ps⁻¹ atom⁻¹.

607 // New text inserted into "Methods (Calculations)"

*The MD simulations were performed for a minimum of 30 ps using a 0.5 fs time-step. The drift in the*
*conserved energy is observed to be below 10^{-6} a.u. ps⁻¹ atom⁻¹.*

**[16] Reviewer #3:** 5. Last but not least (but minor). In Fig. 2f, I can understand the σ -symmetry
states (where they are located). At the same time, I am not able to see where the π ones are
located in the big picture: the inset in my opinion is too small, I would enlarge it or otherwise find
a way to better show the π state positioning in the structure. To which Br/A-cation atoms in the
main picture do they refer to?

**[16] Response:** We thank Reviewer #3 for raising this issue. We have created a new subfigure 2f which
addresses the concerns.

**Figure 2. Experimental and calculated Br K-edge XAS spectra of FAPB, MAPB and CsPB.** **a** Comparison
 of the Total Fluorescence Yield X-ray Absorption Spectroscopy (TFY-XAS, lines) and High Energy
 Resolution Fluorescence Detected X-ray Absorption Spectroscopy (HERFD-XAS, symbols) spectra of
 PbBr_2 (green), FAPB (blue), MAPB (black) and CsPB (red). The vertical dashed lines are guides to the eye
 for features of interest. **b** Comparison of the HERFD-XAS main-edge spectra of the APB compounds. The
 absorption onsets are quantified with sigmoid fits and the main-edge widths are estimated (see text). **c**–
 **e** Experimental versus calculated Br K XAS spectra for FAPB, MAPB and CsPB. The total calculated
 spectrum is shown along with its constituent distributions. The sigma- and pi-symmetry distributions of
 states are denoted as σ , π_1 and π_2 . **f** Crystal structure of MAPB, shown with the molecular orbitals
 associated with σ - and π -symmetry states probed with XAS. The left panel shows the σ orbitals (in cyan
 and yellow) emanating from the Br atom (in red). The right panel shows the π orbitals (in cyan and

yellow) emanating from the Br atom (in red). The lobes of the π orbitals point towards the MA⁺
molecule; the pointing is highlighted by the magenta arrows.

**[17] Minor changes not requested by any reviewer**

We have changed “ σ - π offset” to “ σ - π splitting” throughout the text. We have introduced additional
subheadings into the Results portion of the manuscript to guide the reader. We have expanded the
abbreviations SN, SF and ST to Supplementary Note, Supplementary Figure and Supplementary Table,
respectively. Throughout the manuscript and SI, we have made minor additions or reformulations to
enhance the clarity of the work for the reader. We have added a Data availability section.

The aforementioned changes are highlighted in light gray.

We have removed the former Supplementary Note 6 in the SI, since the content was expanded to
address a Reviewer question above (response [3]) and placed into the Discussion.

REVIEWERS' COMMENTS

Reviewer #2 (Remarks to the Author):

The authors have prepared an excellent response to the concerns raised by myself and the other reviewers. I am satisfied that this paper can be accepted for publication.

Reviewer #3 (Remarks to the Author):

My excuse for the delay.

Text inserted into the manuscript and Supplementary Information, to address reviewer
questions, has been highlighted yellow, green and blue for Reviewers #1, #2 and #3. Minor
additions and reformulations that enhance the clarity of the manuscript are highlighted in gray.

**[1] Reviewer #2:** The authors have prepared an excellent response to the concerns raised by
myself and the other reviewers. I am satisfied that this paper can be accepted for publication.

**[1] Response:** We thank Reviewer #2 for their compliment, detailed assessment of our work and
stimulating comments.

**[2] Reviewer #3:** Authors have provided reasonable answers to the questions I have raised. I
understand that the core of the paper is experimental. Supporting experiments with theoretical
data is a plus for a scientific paper. Anyway, in my opinion standard DFT tends to (as well-
known) underestimate the Conduction Band and its properties. Hybrid and/or GW would
definitely provide a better description of the physics of such (and other) systems.

**[2] Response:** We thank Reviewer #3 for the positive evaluation, stimulating comments and guidance
on improving the study and presentation.

**[3] Reviewer #3:** Having said that, checking the response given by the authors I have a couple
of additional comments.

Response [12]: "DFT, using the same exchange-correlation functional as what we have used
(Perdew-Burke-Ernzerhof (PBE) generalized gradient approximation (GGA)), has been
demonstrated to be reasonably accurate at modeling the conduction band electronic structure of
MAPI, a closely related compound (Yang et al., JPCL 12, 3773 (2021))."

I am fine with this, but I think it should be due to cancellation of errors effects.

**[3] Response:** We agree with Reviewer #3's assessment. We have further clarified the above point by
adding some text, highlighted in blue, to Supplementary Note 3. The full, revised Supplementary Note 3
is as follows.

Supplementary Note 3. **Potential DFT-related limitations with the calculation of the conduction band
electronic structure.**

The common exchange-correlation functionals used in DFT are known to yield underestimated band-
gaps. However, the occupied and unoccupied parts of the computed DOS are generally in good
agreement with experimental photoelectron spectra, stretched by a few percent (Endres *et al.*, JPCL 7,
2722 (2016)). Within the conduction band region, systematic DFT underestimation of a few percent or ~
0.1 eV is not expected to affect the σ - π splitting trend as the magnitude of the splitting is ~ 4 eV.

DFT, using the same exchange-correlation functional as what we have used (Perdew-Burke-Ernzerhof
(PBE) generalized gradient approximation (GGA)), has been demonstrated to be reasonably accurate at
modeling the conduction band electronic structure of MAPI, a closely related compound (Yang *et al.*,
JPCl 12, 3773 (2021)). While there is overall agreement between the calculated conduction band
dispersions and the measured bands (obtained via angle-resolved inverse photoelectron spectroscopy)
over a ~3 eV region of the conduction band, the agreement is not perfect at the 0.1 eV order-of-
magnitude energy scale. We also notice that the good performance of the PBE approximation for lead-
based halide perovskites is related to a favorable error cancellation (e.g. Mosconi *et al.*, JPCC 117, 13902
(2013), Das *et al.*, JPCC 126, 2184 (2022)). It has been reported in the literature that the performance of
standard GGA is good for a number of response properties (e.g. Drisdell *et al.*, ACS Energy Lett. 2, 1183
(2017)).

**[4] Reviewer #3:** Response [13] : " Now we have calculated the total electronic density of states
with and without spin-orbit coupling (SOC) for all three lead bromide perovskites using DFT at
the GGA level (using the Quantum Espresso package). We find that the influence of SOC is
almost negligible in the conduction band region and is very small in the valence band region (0
to -10 eV)."

Probably I do not get the point, but how is it possible that SOC have no impact in the CBM and
in the VBM when calculating total DOS (compared with no SOC)? Bandgaps at the two levels of
theory massively change. (Leppert *et al.*, Phys. Rev. Mater. 3, 103803, 2019). Do authors
maybe refer to higher/lower (in energy) regions in the two bands. At least the bandedges
(mostly the bottom part of the conduction band) should be dramatically affected.

Maybe it is my misinterpretation of the authors reply, and accordingly I would authors to confirm
that I am wrong

**[4] Response:** We feel sorry for the lack of clarity in our response. As Reviewer #3 has guessed, our
statement regarding almost negligible SOC influence is intended to refer to the higher/lower (in energy)
regions of the two bands. What we meant to say is the following.

We find that the influence of SOC, considering our convolution scheme, on the peak positions (varies
less than 0.05 eV) as well as the orbital characters of the states in the conduction band region, at higher
energy with respect to the CBM, is negligible and is very small in the valence band region (0 to -10 eV,
except for Cs 5*p* in CsPbBr₃). We primarily focused on the influence of SOC effects on the important
states corresponding to the dominant peaks in the experimental spectra. The green arrows in the figure
below highlight these states. As Reviewer #3 has pointed out, there is a clear deviation in the onset of
TDOS (corresponds to the CBM) between the TDOS with and without SOC. The brown arrows in the
figure below highlight this.

Additionally, we note that the effect of SOC on density of states is similar for the three lead bromide
perovskites and here we focus on relative, A-cation-induced changes in the σ - π energy separation
between the three perovskites. From this standpoint, the effect of SOC is expected to be negligible.

We have revised the text in the section titled "Calculated ground-state electronic structure", where the
new/revised text is highlighted in blue.

79 // Revised paragraph in the section **Calculated ground-state electronic structure**

To evaluate the possible influence of spin-orbit coupling (SOC) on the conclusions of our analysis, we
 performed total electronic density of states (TDOS) calculations for all three lead bromide perovskites
 using DFT with/without SOC effects. Further details are found in the Methods. The results of the
 calculations, shown in Supplementary Fig. 5, suggest that given our convolution scheme the influence of
 SOC on the peak positions (variations of less than 0.05 eV) at higher energy with respect to the CBM as
 well as the orbital characters of the states in the conduction band region is negligible. Hence, the
 influence of SOC on the analysis of the Br *K*-edge XAS data is negligible also. Furthermore, we note that
 the SOC effect on the lead and bromine PDOS of the three perovskites should be similar, given the same
 PbBr_6 octahedra, and in this work we focus on relative, A-cation-induced changes in the σ - π energy
 separation between the three perovskites. The influence of SOC is very small for the valence band
 region up to -10 eV, except for the Cs *5p* feature in CsPB (-6 to -10 eV). The computed TDOS shows the
 Pb *5d* splitting around -12 to -20 eV, which is consistent with the shallow core photoelectron spectra of
 Pb *5d* (Supplementary Fig. 6) and confirms the inclusion of SOC in the TDOS calculations.

[5] Reviewer #3: Other answers seem quite reasonable to me.

[5] Response: We thank Reviewer #3 for the careful review.